# Endothelial ZEB1 promotes angiogenesis-dependent bone formation and reverses osteoporosis

Rong Fu[1,5], Wen-Cong Lv[1,5], Ying Xu[1,5], Mu-Yun Gong[1,5], Xiao-Jie Chen[2], Nan Jiang[1], Yan Xu[3], Qing-Qiang Yao[3], Lei Di[1], Tao Lu[4], Li-Ming Wang[3], Ran Mo[2] & Zhao-Qiu Wu[1]*

Recent interest in the control of bone metabolism has focused on a specialized subset of $CD31^{hi}endomucin^{hi}$ vessels, which are reported to couple angiogenesis with osteogenesis. However, the underlying mechanisms that link these processes together remain largely undefined. Here we show that the zinc-finger transcription factor ZEB1 is predominantly expressed in $CD31^{hi}endomucin^{hi}$ endothelium in human and mouse bone. Endothelial cell-specific deletion of ZEB1 in mice impairs $CD31^{hi}endomucin^{hi}$ vessel formation in the bone, resulting in reduced osteogenesis. Mechanistically, ZEB1 deletion reduces histone acetylation on *Dll*4 and *Notch1* promoters, thereby epigenetically suppressing Notch signaling, a critical pathway that controls bone angiogenesis and osteogenesis. ZEB1 expression in skeletal endothelium declines in osteoporotic mice and humans. Administration of *Zeb1*-packaged liposomes in osteoporotic mice restores impaired Notch activity in skeletal endothelium, thereby promoting angiogenesis-dependent osteogenesis and ameliorating bone loss. Pharmacological reversal of the low ZEB1/Notch signaling may exert therapeutic benefit in osteoporotic patients by promoting angiogenesis-dependent bone formation.

[1] State Key Laboratory of Natural Medicines, School of Basic Medicine and Clinical Pharmacy, China Pharmaceutical University, Nanjing 211198, China. [2] State Key Laboratory of Natural Medicines, Jiangsu Key Laboratory of Drug Discovery for Metabolic Diseases, Center of Advanced Pharmaceuticals and Biomaterials, China Pharmaceutical University, Nanjing 210009, China. [3] Department of Orthopedic Surgery, Digital Medicine Institute, The Affiliated Nanjing Hospital of Nanjing Medical University, Nanjing 210006, China. [4] State Key Laboratory of Natural Medicines, Laboratory of Molecular Design and Drug Discovery, School of Science, China Pharmaceutical University, Nanjing 211198, China. [5]These authors contributed equally: Rong Fu, Wen-Cong Lv, Ying Xu, Mu-Yun Gong. *email: zqwu@cpu.edu.cn

A specific vessel subtype, strongly positive for CD31 and endomucin (CD31$^{hi}$EMCN$^{hi}$), has been recently reported to couple angiogenesis with osteogenesis in the bone[1–4]. CD31$^{hi}$EMCN$^{hi}$ vessels, residing in the bone marrow near the growth plate, represent the key component of a metabolically specialized bone microenvironment with privileged access to nutrients and oxygen, thereby promoting the growth potential of surrounding perivascular cells, including pre-osteoprogenitors, osteoprogenitors, mature osteoblasts, mature and hypertrophic chondrocytes, and hematopoietic stem cells[1–5]. It is increasingly appreciated that skeletal CD31$^{hi}$EMCN$^{hi}$ vessel numbers decline in ageing and ovariectomy (OVX)-induced osteoporotic mice and in osteoporotic patients[1,4,6–9]. Osteoporosis is a systemic bone disease characterized by abnormal bone metabolism, low bone mass and density, impaired bone microstructure, and increased bone fragility and fracture prone[6–9]. Reduced CD31$^{hi}$EMCN$^{hi}$ endothelium levels in the bone are strongly associated with osteoporosis progression. Thus, pharmacological restorage of CD31$^{hi}$EMCN$^{hi}$ endothelium levels in the bone may exert therapeutic benefit in osteoporotic patients by promoting angiogenesis-dependent osteogenesis. The evolutionarily conserved Dll4/Notch signaling in the bone endothelium has been reported to increase CD31$^{hi}$EMCN$^{hi}$ endothelium levels and thus promote bone formation by upregulating secretion levels of Noggin, a Notch-controlled angiocrine factor[2]. However, it is currently unclear how Notch activity is tightly controlled in the bone endothelium and whether declined endothelial Notch signaling is linked to osteoporosis.

Zinc-finger E-box-binding homeobox 1 (ZEB1) is most frequently characterized as a zinc-finger transcription factor that triggers the epithelial–mesenchymal transition (EMT) programs critical to development as well as a range of pathologic states, including cancer[10–13]. Previous studies have demonstrated that global deletion of ZEB1 in mice elicits perinatal lethality due to severely skeletal defects[14]. However, the primary cell lineage(s) affected in these mice and the underlying molecular mechanisms remain undefined. In addition, it remains unclear whether ZEB1 is required for postnatal bone formation or whether ZEB1 loss in certain cell lineage(s) is linked to osteoporotic disease in human and mice. Here we show that ZEB1 is predominantly expressed in CD31$^{hi}$EMCN$^{hi}$ endothelium in human and mouse bone, and ZEB1 levels decline in ageing and OVX-induced osteoporotic mice, as well as in osteoporotic patients. Endothelial cell (EC)-specific deletion of ZEB1 in mice reduces CD31$^{hi}$EMCN$^{hi}$ vascularization and osteogenesis by epigenetically repressing Dll4/Notch signaling. Administration of recombinant Dll4 protein efficiently restores CD31$^{hi}$EMCN$^{hi}$ vessel growth and bone formation in ZEB1-deleted mice. Importantly, we observe low ZEB1-Dll4/Notch signaling in OVX-induced osteoporotic mice and in vivo ZEB1 gene delivery restores impaired Notch signaling, thereby boosting CD31$^{hi}$EMCN$^{hi}$ vessel formation, promoting osteogenesis, and ameliorating bone loss in OVX-induced osteoporotic mice. In summary, our results lay the foundation for new therapeutic strategies in osteoporosis treatment by promoting angiogenesis-dependent bone formation.

## Results

### ZEB1 is predominantly expressed in CD31$^{hi}$EMCN$^{hi}$ bone ECs.
We scanned tissues that were collected from juvenile 3-week-old mice for ZEB1 protein expression. Interestingly, we found that ZEB1, as detected by immunofluorescence, was expressed in the endothelium of skeletal elements such as the tibia, sternum, and vertebra at significantly higher positivity and expression levels than in the endothelium of non-skeletal organs such as the spleen, lung, kidney, liver, and heart (Fig. 1a, b). Further, we observed

that ZEB1 protein was predominantly expressed in metaphyseal CD31$^{hi}$EMCN$^{hi}$ (termed as type H) endothelium of tibia, while it was essentially undetectable in the CD31$^{low}$EMCN$^{low}$ (termed as type L) endothelium found within the bone marrow (Fig. 1c, d). These observations suggest a markedly distinct ZEB1 expression pattern between type H and L vessels in mouse long bone (e.g., tibia), a finding that could be extended to other skeletal elements such as the sternum, calvarium, and vertebra (Fig. 1c, d). Importantly, the distinct ZEB1 expression pattern observed in mouse bone was also presented in human tibia (Fig. 1c, d). Furthermore, we performed a quantitative reverse transcription PCR (RT-qPCR) assay on fluorescence-activated cell (FACS)-sorted type H vs. type L tibial ECs of 3-week-old mice. The results demonstrated that Zeb1 transcript levels in type H tibial ECs were also significantly higher than in type L ECs (Fig. 1e). Intriguingly, transcript levels of Zeb2, another member of ZEB family, in type H bone ECs were remarkably lower than in type L ECs, an expression pattern that is highly distinct from ZEB1 in bone ECs (Fig. 1e). It has been previously reported that type H vessel numbers and bone mass decline in tandem with ageing in human and murine skeletal system[1,6] (Fig. 1f–h). Notably, ZEB1 protein in type H tibial ECs was remarkably more abundant in juvenile (3-week-old) mice relative to (10-week-old) adults and was largely absent in aged (50-week-old) animals (Fig. 1h, i), suggesting that ZEB1 protein is dynamically controlled during ageing. On the basis of the specific expression pattern of ZEB1 in the skeletal system, we considered the possibility that ZEB1 expressed by type H bone ECs functions as a key signaling regulator of angiogenesis and bone formation.

### Endothelial ZEB1 deletion reduces type H vessel formation.
To investigate ZEB1 function in the bone endothelium, conditional Zeb1$^{floxed/floxed}$ (Zeb1$^{fl/fl}$) mice were generated in our laboratory (Supplementary Fig. 1a; ref. [15]) and crossed with Tie2-Cre[16,17] transgenic line to generate Tie2-Cre$^-$;Zeb1$^{fl/fl}$ control and Tie2-Cre$^+$;Zeb1$^{fl/fl}$ EC-specific ZEB1 knockout mice (designated Zeb1$^{WT}$ and Zeb1$^{\Delta EC}$ mice, respectively). In addition, Zeb1$^{fl/fl}$ mice were also mated with Cdh5(PAC)-CreERT2[18] transgenic line and the resulting Cdh5(PAC)-CreERT2$^-$;Zeb1$^{fl/fl}$ and Cdh5(PAC)-CreERT2$^+$;Zeb1$^{fl/fl}$ mice were treated with tamoxifen to generate control and EC-specific ZEB1 knockout mice (designated Zeb1$^{WT}$ and Zeb1$^{i\Delta EC}$ mice, respectively). RT-qPCR analysis of FACS-sorted tibial ECs revealed significantly reduced Zeb1 transcript levels in 3-week-old Zeb1$^{\Delta EC}$ compared with control littermates (Supplementary Fig. 1b, c). Also, Zeb1 transcript levels were remarkably decreased in tibial ECs of 3-week-old Zeb1$^{i\Delta EC}$ mice that were intraperitoneally (i.p.) injected with 0.1 mg tamoxifen every day at postnatal day 8 (P8) for 7 consecutive days, as compared with Zeb1$^{WT}$ control mice receiving identical tamoxifen treatment (Supplementary Fig. 1c). Furthermore, immunofluorescence analysis of tibial sections revealed that ZEB1 protein was efficiently depleted in type H bone ECs but not perivascular cells of 3-week-old Zeb1$^{\Delta EC}$ mice (Supplementary Fig. 1d, e) and Zeb1$^{i\Delta EC}$ mice (Supplementary Fig. 1f, g). Constitutive and tamoxifen inducible inactivation of endothelial ZEB1 in 3-week-old mice markedly decreased the density of type H vessels, but not type L vessels, in skeletal elements such as tibia, sternum, vertebra, and calvarium, as assessed by immunofluorescence (Fig. 2a–c and Supplementary Fig. 2a, b) and FACS (Fig. 2d, e) analyses. Similarly, 10-week-old adult Zeb1$^{\Delta EC}$ mice and Zeb1$^{i\Delta EC}$ mice that were i.p. injected with 1.0 mg tamoxifen every other day at 7 weeks of ages for 2 consecutive weeks both exhibited substantially reduced type H but not type L vessel density compared with their corresponding littermate controls (Supplementary Fig. 2c-f). By contrast, Zeb1$^{\Delta EC}$ mice exhibited comparable CD31$^+$ vessel densities in non-skeletal tissues such as the heart,

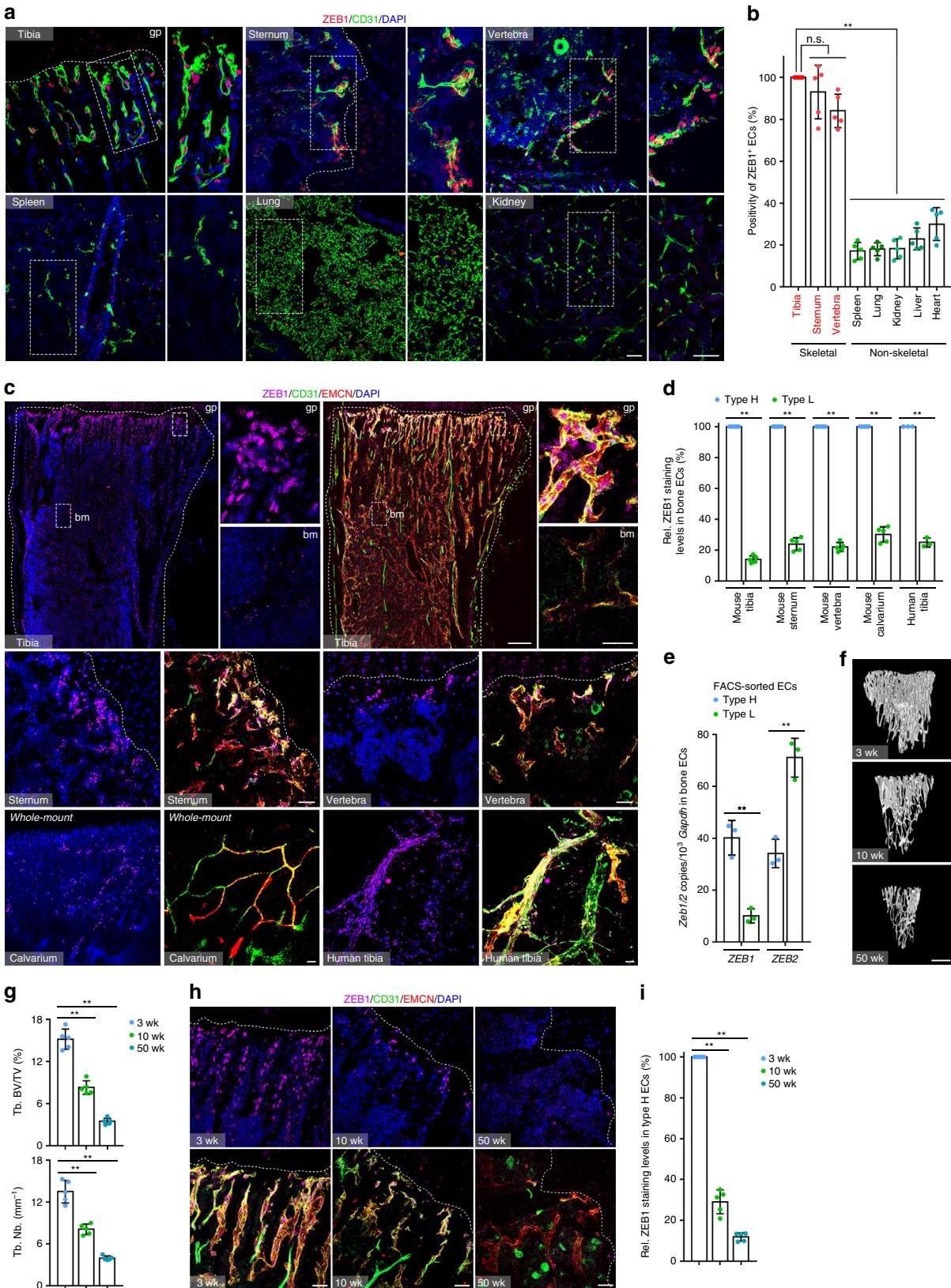

lung, liver, kidney, and spleen relative to littermate controls (Supplementary Fig. 2g, h). Vascular endothelial growth factor A (VEGFA), a potent proangiogenic growth factor secreted by endothelial column/arches, perivascular cells, and mature/hypertrophic chondrocytes[19], was robustly decreased in the tibia of 3-week-old $Zeb1^{\Delta EC}$ mice, as assessed by immunofluorescence

analysis (Fig. 2f, g). RT-qPCR analysis of FACS-sorted tibial ECs also revealed a strong reduction in $Vegfa$ transcript levels in 3-week-old $Zeb1^{\Delta EC}$ mice compared with littermate controls (Fig. 2h). These findings are consistent with previous reports demonstrating that ZEB1 is positively associated with VEGFA expression in cancer cells[20] and cancer-associated fibroblasts[15].

**Fig. 1 ZEB1 is primarily expressed in type H vessels of human and murine bone. a** Representative confocal images showing ZEB1 expression in $CD31^+$ ECs of the tibia, sternum, vertebra, spleen, lung, and kidney dissected from juvenile 3-week-old mice ($n = 5$). Nuclei, 4′,6-diamidino-2-phenylindole (DAPI). Magnified areas of dashed boxed sections are shown in right panels. gp growth plate. Scale bar, 30 μm. **b** Quantification of positivity of $ZEB1^+$ ECs as shown in **a** ($n = 5$ independent experiments). **c** Comparisons of ZEB1 staining levels in $CD31^{hi}EMCN^{hi}$ (type H) vs. $CD31^{lo}EMCN^{lo}$ (type L) ECs of 3-week-old tibia, sternum, vertebra, and calvarium ($n = 5$), and human tibia ($n = 3$). Magnified areas of dashed boxed sections in top row are shown in right panels. bm bone marrow, EMCN endomucin. Scale bar: 30 μm, 200 μm (top row, lower magnification images). **d** Quantification of ZEB1 staining levels in two types of bone ECs as shown in **c** ($n = 5$ or 3 independent experiments). **e** RT-qPCR analysis of *Zeb1* and *Zeb2* transcripts in FACS-sorted type H and type L bone ECs of 3-week-old mice ($n = 3$ independent experiments). **f** Representative micro-CT images of trabecular bone of 3-, 10-, and 50-week-old tibia ($n = 5$). Scale bar, 0.2 mm. **g** Quantification of trabecular bone fraction (Tb. BV/TV; trabecular bone, bone volume/tissue volume) and trabecular number (Tb. Nb.) in tibia as shown in **f** ($n = 5$ independent experiments). **h** Representative confocal images of ZEB1, CD31, and EMCN immunostained sections of 3-, 10- or 50-week-old tibia ($n = 5$, each). Scale bar, 30 μm. **i** Quantification of ZEB1 staining levels in tibial type H ECs as shown in **h** ($n = 5$ independent experiments). All data are represented as mean ± SD. **$P < 0.01$; NS not significant. Differences are tested using one-way ANOVA with Tukey's post hoc test (**b**, **g**, **i**) and unpaired two-tailed Student's *t*-test (**d**, **e**). The source data are provided as a Source Data file.

Microphil-perfused angiography demonstrated a markedly reduced vessel number and volume in the metaphysis of 3-week-old $Zeb1^{\Delta EC}$ tibia (Fig. 2i) where disrupted column/arch patterning and impaired filopodia extension were observed (Fig. 2j, k).

**Endothelial ZEB1 deletion impairs bone formation.** Severely defective angiogenesis in the skeletal organs of $Zeb1^{\Delta EC}$ mice at 3, 10, and 50 weeks of ages was accompanied by significantly reduced body weight and shortened femoral length (Fig. 3a and Supplementary Fig. 3a). Micro-computed tomography (micro-CT) and histo-morphometric analyses revealed significant loss of trabecular bone density (trabecular bone volume per tissue volume (Tb. BV/TV)), trabecular number (Tb. Nb.), trabecular thickness (Tb. Th.), and cortical thickness (Ct. Th.) in the tibia of 3-, 10-, and 50-week-old $Zeb1^{\Delta EC}$ mice (Fig. 3b, c and Supplementary Fig. 3b–d). By contrast, no structural alterations were detected in 3-week-old $Zeb1^{\Delta EC}$ non-skeletal organs such as the heart, liver, spleen, lung, and kidney (Supplementary Fig. 3e). Notably, 3-week-old $Zeb1^{i\Delta EC}$ mice also exhibited markedly reduced body weight (Supplementary Fig. 3f, left two columns) in tandem with severely impaired bone formation (Supplementary Fig. 3g, h). As body weight is believed to have a profound impact on bone mass, we further assessed whether impaired bone formation observed in endothelial ZEB1-deleted mice was due to reduced body weight of the mice. To this end, we induced endothelial ZEB1 deletion in mice at 7 weeks of ages, a stage that skeletal growth has been completed. As shown, body weight of 10-week-old adult $Zeb1^{WT}$ and $Zeb1^{i\Delta EC}$ mice was largely comparable (Supplementary Fig. 3f, right two columns), whereas bone mass of $Zeb1^{i\Delta EC}$ mice was significantly lower than $Zeb1^{WT}$ mice, as assessed by micro-CT analysis (Supplementary Fig. 3i, j). Thus, these findings demonstrate that endothelial ZEB1 deletion at young or adult ages both elicits impairment of bone formation independently of body weight reduction in the mice. Calcein double labeling further confirmed strongly reduced bone formation rates in the tibia of 3-week-old $Zeb1^{\Delta EC}$ mice (Fig. 2d, e). Importantly, 3-week-old $Zeb1^{\Delta EC}$ and $Zeb1^{i\Delta EC}$ mice showed significantly reduced numbers of $Runx2^+$ pre-osteoprogenitors[21], $Osterix^+$ osteoprogenitors[22], as well as $ALP^+$ mature osteoblasts[23] in tandem with impaired mineralized bone mass (Alizarin staining), but without appreciable changes in the $TRAP^+$ preosteoclasts numbers[24] (Fig. 3f–l and Supplementary Fig. 4a–d). Consistent with the changes observed in 3-week-old juvenile bone, substantially reduced numbers of $Runx2^+$ pre-osteoprogenitors and $Osterix^+$ osteoprogenitors were also detected in 10-week-old adult $Zeb1^{\Delta EC}$ tibia (Supplementary Fig. 4e, f). Similarly, no considerable change in the $TRAP^+$ osteoclasts numbers was observed in the tibia of 10-week-old $Zeb1^{\Delta EC}$ mice compared with $Zeb1^{\Delta EC}$ mice (Supplementary Fig. 4g, h).

**ZEB1 depletion represses Dll4/Notch signaling in bone ECs.** To investigate the potential mechanisms by which endothelial ZEB1 regulates bone angiogenesis and osteogenesis, bone ECs of $Zeb1^{WT}$ and $Zeb1^{\Delta EC}$ mice were freshly isolated and subjected to RT-qPCR screening analysis. As shown, transcript levels of *Dll4*, *Notch1*, *Notch2*, *Notch3*, *Notch4*, *Hes1*, *Hes5*, *Hey1*, *Heyl*, and *Efnb2* were markedly reduced in $Zeb1^{\Delta EC}$ mice, suggesting that the Notch signaling was downregulated in the $Zeb1^{\Delta EC}$ bone ECs (Fig. 4a). Intriguingly, *Zeb2* transcript levels were also remarkably decreased in $Zeb1^{\Delta EC}$ bone ECs (Fig. 4a), suggesting that ZEB1 excision in bone ECs does not induce a compensatory increase in *Zeb2* expression but markedly reduces its expression. Further, immunofluorescence analysis revealed that expression levels of Dll4 and Notch1 in $EMCN^+$ bone ECs were markedly decreased in the tibia of $Zeb1^{\Delta EC}$ and $Zeb1^{i\Delta EC}$ mice relative to their corresponding littermate controls (Fig. 4b, e). To assess whether ZEB1 controls Notch activity in an EC-autonomous manner, we performed a RT-qPCR assay on the in vitro cultured bone ECs that were isolated from $Zeb1^{fl/fl}$ tibia and infected with adeno-βGal or adeno-Cre (to generate control and ZEB1-deleted cells, respectively). The results showed a significant reduction in transcript levels of *Zeb1*, *Zeb2*, and key components of Dll4/Notch pathway in ZEB1-deleted bone ECs relative to control cells (Fig. 4f, g). Protein levels of Dll4, Notch1, and the intracellular domain of Notch1, NICD1 (which is the active form of Notch1), were strongly decreased in ZEB1-deleted bone ECs compared with control cells, as assessed by immunoblot and immunofluorescence analyses (Fig. 4h, i).

Similar to a well-established zinc-finger transcription factor Snail1[25–27], ZEB1 acts as a transcriptional repressor/activator by directly binding to E-box elements (consensus sequence, CANNTG), which are located within the promoter regions of target genes[28,29]. In this regard, initial searches identified multiple conserved E-box elements within the proximal regions of the murine *Dll4* and *Notch1* promoters (Fig. 5a). Indeed, ZEB1-deleted bone ECs markedly decreased luciferase activity of wild-type (WT) *Dll4* and *Notch1* reporter constructs, without affecting luciferase activity of mutant (MUT) *Dll4* and *Notch1* reporter constructs harboring successive point mutations in all E-boxes (Fig. 5b). In control bone ECs, luciferase activity of MUT *Dll4* and *Notch1* reporter constructs was significantly lower than corresponding WT reporter constructs, confirming that ZEB1 functions as a transcription activator of *Dll4* and *Notch1* genes (Fig. 5b). Conversely, exogenous ZEB1-expressing bone ECs remarkably increased luciferase activity of WT *Dll4* and *Notch1* reporter constructs, wereas it failed to affect luciferase activity of MUT *Dll4* and *Notch1* reporter constructs (Fig. 5c). After chromatin immunoprecipitation (ChIP) of endogenous ZEB1 in bone ECs followed by PCR of the proximal and distal regions of the murine *Dll4* and *Notch1* promoters, ZEB1-ChIPs were found

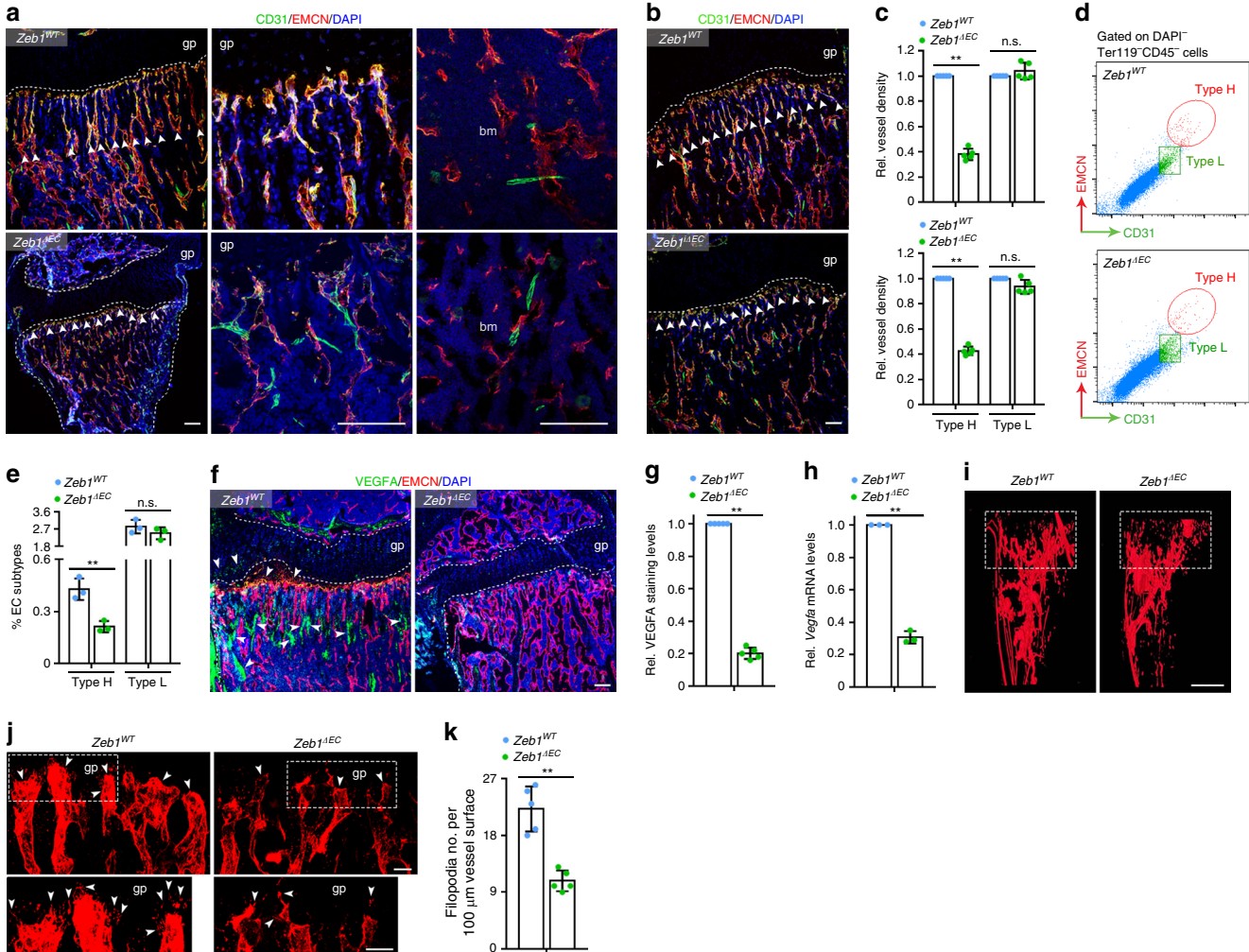

**Fig. 2 EC-specific deletion of ZEB1 reduces bone angiogenesis. a, b** Representative confocal images of CD31/EMCN immunostaining in tibia of juvenile 3-week-old $Zeb1^{\Delta EC}$ mice (**a**; $n = 5$) and $Zeb1^{i\Delta EC}$ mice (**b**; $n = 5$, each), and their corresponding littermate controls ($n = 5$, each). Magnified images of type H and type L vessels are shown in middle and right panels (**a**), respectively. Scale bar, 100 μm. Arrowhead denotes interface between type H and L vessels. **c** Quantification of type H and type L vessel densities in tibia as described in **a** and **b** ($n = 5$ independent experiments). **d** FACS plots of CD31 and EMCN double-stained single-cell suspensions from 3-week-old $Zeb1^{WT}$ and $Zeb1^{\Delta EC}$ tibia ($n = 5$, each). **e** Quantification of type H and type L bone ECs in LIN⁻ (i.e., DAPI⁻Ter119⁻CD45⁻) cells as shown in **d** ($n = 5$ independent experiments). **f** Confocal images of 3-week-old $Zeb1^{WT}$ and $Zeb1^{\Delta EC}$ tibia ($n = 5$, each) immunostained with VEGFA and EMCN. Scale bar, 100 μm. Arrowhead marks VEGFA positively staining in endothelial column/arches, surrounding perivascular cells, and mature/hypertrophic chondrocytes. **g** Quantification of VEGFA staining levels in tibia as shown in **f** ($n = 5$ independent experiments). **h** RT-qPCR analysis of *Vegfa* transcript in FACS-sorted bone ECs of 3-week-old $Zeb1^{WT}$ and $Zeb1^{\Delta EC}$ mice ($n = 3$ independent experiments). **i** Representative angiographic images of vascular organization of 3-week-old $Zeb1^{WT}$ and $Zeb1^{\Delta EC}$ tibia ($n = 5$, each). Dashed boxed sections denote vessels near metaphysis. Scale bar, 0.5 mm. **j** Confocal images of EMCN immunostained distal columns and arches next to gp in 3-week-old $Zeb1^{WT}$ and $Zeb1^{\Delta EC}$ tibia ($n = 5$, each). Arrowhead denotes filopodia extension. Magnified areas of dashed boxed sections are shown in bottom panels. Scale bar, 20 μm. **k** Quantification of filopodia numbers in tibia as shown in **j** ($n = 5$ independent experiments). All data are represented as mean ± SD. **$P < 0.01$; NS not significant. Differences are tested using unpaired two-tailed Student's *t*-test. The source data are provided as a Source Data file.

to be enriched in the proximal, but not distal, regions of both promoters, suggesting that ZEB1 activated *Dll4* and *Notch1* transcription by specifically binding to promoter proximal regions (Fig. 5d, e). The ability of zinc-finger transcription factors (e.g., Snail) to transactivate target genes has been linked to the recruitment of histone-modifying cofactors to the promoters of target genes, which can specifically enhance histone acetylation but not methylation on the promoters[25,30]. ChIP-qPCR analysis showed that ZEB1-deleted bone ECs significantly reduced histone acetylation, including histone H3 lysine 4 acetylation (H3K4Ac), H3K14Ac, and H3K18Ac, on the promoters of *Dll4* and *Notch1*, without affecting recruitment of histone H3 lysine 27 trimethylation (H3K27me3), a reported repressive histone marker[25,31].

(Fig. 5f). To uncover the cofactors that are responsible for histone acetylation on the promoters of *Dll4* and *Notch1*, we performed co-IP experiments in bone ECs. The results demonstrated that ZEB1 was associated with histone acetyltransferases (HATs), CREB-binding protein (CBP)/p300 in bone ECs (Fig. 5g). Sequential ChIP analyses revealed that ZEB1 and CBP co-occupied *Dll4* and *Notch1* promoters in bone ECs (Fig. 5h). A promoter luciferase reporter assay was then performed in 293T cells that were transiently transfected with HA-ZEB1, HA-p300, or HA-CBP alone, or ZEB1 in combination with p300 or CBP (Fig. 5i). The results showed that overexpression of ZEB1, p300, or CBP alone increased *Dll4* (*Notch1*) promoter activity by 88% (75%), 20% (16%), or 23% (15%), respectively, whereas

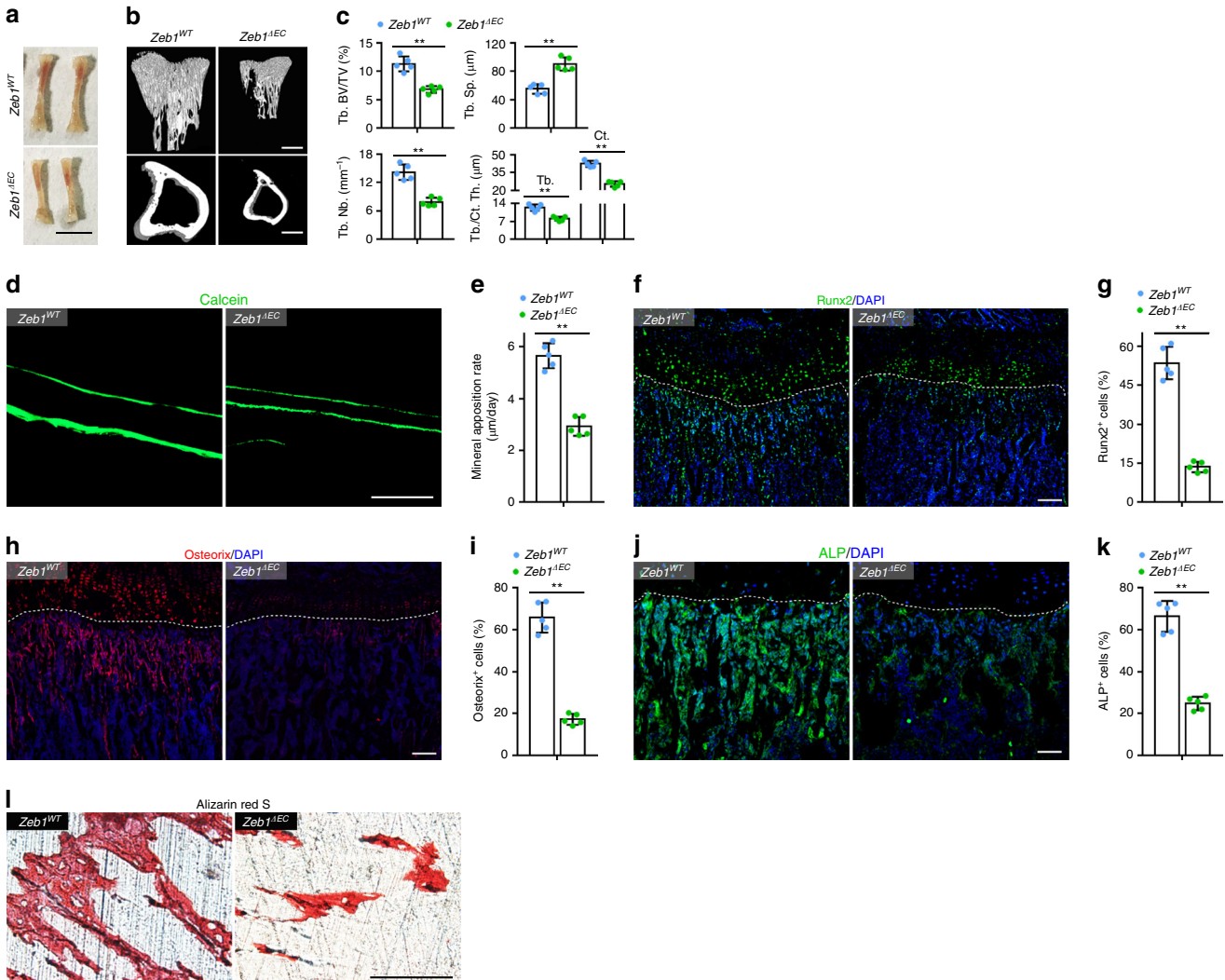

**Fig. 3 EC-specific deletion of ZEB1 impairs angiogenesis-dependent bone formation. a** Representative images of freshly dissected tibia of 3-week-old *Zeb1^WT^* and *Zeb1^ΔEC^* mice (*n* = 5, each). Scale bar, 50 mm. **b** Representative micro-CT images of trabecular bone (top panels) and cortical bone (bottom panels) of 3-week-old *Zeb1^WT^* and *Zeb1^ΔEC^* mice (*n* = 5, each). Scale bar, 0.2 mm. **c** Quantification of Tb. BV/TV trabecular separation (Tb. Sp.), Tb. Nb. trabecular thickness (Tb. Th.), cortical thickness (Ct. Th.) in the tibia as shown in **b** (*n* = 5 independent experiments). **d** Representative images of calcein double labeling of *Zeb1^WT^* and *Zeb1^ΔEC^* tibia (*n* = 5, each). Scale bar, 100 μm. **e** Quantification of dynamic bone formation in tibia as shown in **d** (*n* = 5 independent experiments). **f**, **h**, **j** Representative images of 3-week-old *Zeb1^WT^* and *Zeb1^ΔEC^* tibia (*n* = 5, each) immunostained with Runx2 (**f**), Osterix (**h**), and ALP (**j**). Scale bar, 100 μm. **g**, **i**, **k** Quantification of Runx2⁺ (**g**), Osterix⁺ (**i**), and ALP⁺ (**k**) cells in the tibia as shown in **f**, **h**, **j** (*n* = 5 independent experiments). **l** Representative images for Alizarin red S staining of 3-week-old *Zeb1^WT^* and *Zeb1^ΔEC^* tibia (*n* = 5, each). Scale bar, 100 μm. All data are represented as mean ± SD. **$P < 0.01$; NS not significant. Differences are tested using unpaired two-tailed Student's *t*-test. The source data are provided as a Source Data file.

overexpression of ZEB1 in combination with p300 or CBP resulted in a 6.0 (4.1)-fold or 7.0 (5.4)-fold increase in *Dll4* (*Notch1*) promoter activity, respectively (Fig. 5j). These findings suggest that ZEB1 interacts with CBP/p300 in bone ECs and they co-occupy the promoters of *Dll4* and *Notch1* to enhance histone acetylation on the promoters, thus inducing transcriptional activation of *Dll4* and *Notch1*.

**Treatment with r.Dll4 protein restores bone formation.** Given these results, we next sought to determine the degree to which Dll4 serves as a ZEB1 target substrate required for angiogenesis and bone formation. Treatment of in vitro cultured ZEB1-deleted bone ECs with recombinant Dll4 (r.Dll4) protein remarkably restored expressions of key components of Dll4/Notch pathway such as *Dll4, Notch1, Hes1, Hey1*, and *Efnb2*, as assessed by RT-qPCR analysis (Supplementary Fig. 5a). Importantly, the impaired

expression of NICD1 in ZEB1-deleted bone ECs was also fully recovered following r.Dll4 treatment, as assessed by immuno-fluorescence and immunoblot analyses (Fig. 6a, b), suggesting that ZEB1-deleted bone ECs with impaired Notch activity were responsive to Dll4 stimulation. Our findings are consistent with previous reports demonstrating that treatment of muscle satellite cells and ECs with r.Dll4 protein robustly stimulates Notch activity[32,33]. Daily injection of *Zeb1^WT^* mice with 1 μg/g r.Dll4 protein for 2 consecutive weeks (starting from P4) before analysis at P21 markedly increased protein expressions of Dll4 and Notch1 in the tibia (Supplementary Fig. 5b), but had no obvious impact on body weight (Supplementary Fig. 5c), bone angiogenesis (Fig. 6c, d), or osteogenesis (Fig. 6e–h). In sharp contrast, administration of *Zeb1^ΔEC^* mice with r.Dll4 protein remarkably restored body weight (Supplementary Fig. 5c), enhanced type H vessel formation and VEGFA expression in the tibia (Fig. 6c, d),

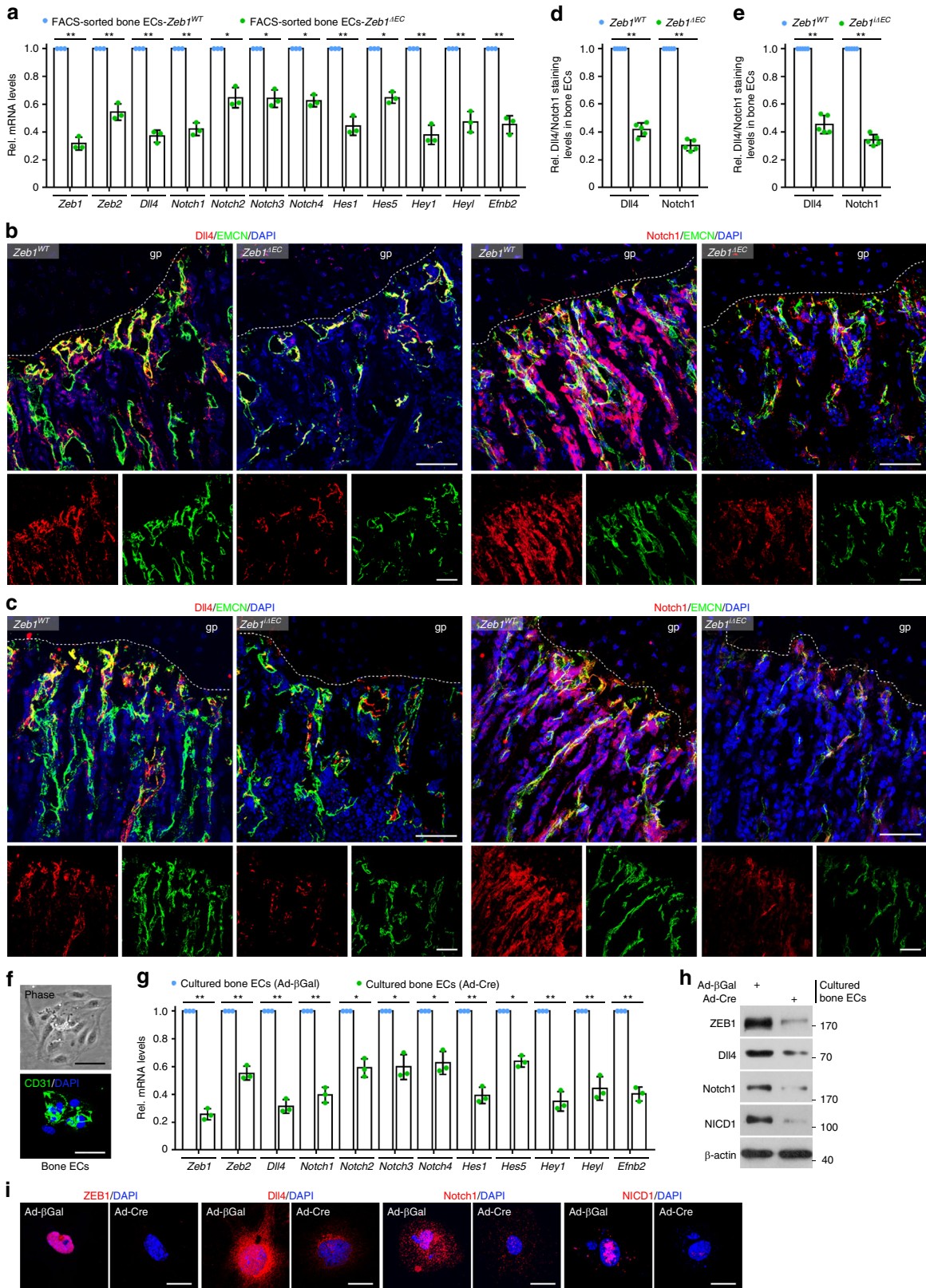

improved bone formation (Fig. 6e, f), and normalized the numbers of Runx2+ pre-osteoprogenitors, Osterix+ osteoprogenitors, and ALP+ mature osteoblasts (Fig. 6g, h). Recently, angiocrine factors secreted by vascular niche have been reported to play critical roles in promoting organ development, tumor growth, and tissue regeneration[34,35]. We therefore sought to identify potential pro-

osteogenic factors that are actively involved in regulation of endothelial Notch-dependent osteogenesis. As shown, *Tgfb1*, *Tgfb2*, *Tgfb3*, *Bmp2*, *Bmp4*, *Pgf* (encoding PLGF), *Fgf1*, and *Nog* (encoding Noggin) transcripts were dramatically reduced in FACS-sorted bone ECs of *Zeb1*$^{\Delta EC}$ mice, among which *Tgfb1*, *Tgfb2*, *Bmp2*, *Bmp4*, *Fgf1*, and *Nog* transcripts were markedly

**Fig. 4 Endothelial ZEB1 inactivation downregulates Notch activity. a** RT-qPCR analysis of *Zeb1, Zeb2, Dll4, Notch1, Notch2, Notch3, Notch4, Hes1, Hes5, Hey1, Heyl*, and *Efnb2* in FACS-sorted bone ECs of 3-week-old *Zeb1$^{WT}$* and *Zeb1$^{ΔEC}$* mice (*n* = 3 independent experiments). **b, c** Representative confocal images of Dll4 (left panels) and Notch1 (right panels) immunostaining in the tibia of 3-week-old *Zeb1$^{ΔEC}$* mice (**b**; *n* = 5) and *Zeb1$^{iΔEC}$* mice (**c**; *n* = 5), and their corresponding littermate controls (*n* = 5, each). Scale bar, 60 μm. **d** Quantification of Dll4 (left panels) and Notch1 (right panels) staining levels in EMCN$^+$ bone ECs of 3-week-old *Zeb1$^{WT}$* and *Zeb1$^{ΔEC}$* mice as described in **b** (*n* = 5 independent experiments). **e** Quantification of Dll4 (left panels) and Notch1 (right panels) staining levels in EMCN$^+$ bone ECs of 3-week-old *Zeb1$^{WT}$* and *Zeb1$^{iΔEC}$* mice as described in **c** (*n* = 5 independent experiments). **f** Representative phase contrast (top panel) and confocal (CD31 immunostaining; bottom panel) images of magnetic activated cell sorting (MACS)-sorted bone ECs. Scale bar, 50 μm. **g** RT-qPCR analysis of *Zeb1* and the indicated Notch pathway components in the in vitro cultured control (i.e., adeno-βGal-infected) and ZEB1-deleted (adeno-Cre-infected) bone ECs (*n* = 3 independent experiments). **h** Immunoblot analysis of ZEB1, Dll4, Notch1, and NICD1 in control and ZEB1-deleted bone ECs as described in **g**. Representative blots as shown are from three independent experiments. **i** Representative confocal images of control and ZEB1-deleted bone ECs with immunostaining of ZEB1, Dll4, Notch1, and NICD1. Scale bar, 20 μm. Images as shown are from three independent experiments. All data are represented as mean ± SD. **P < 0.01, *P < 0.05. Differences are tested using one-way ANOVA with Tukey's post hoc test (**a, g**) and unpaired two-tailed Student's *t*-test (**d, e**). The source data are provided as a Source Data file. Unprocessed original scans of blots are shown in Source Data file.

restored in bone ECs of *Zeb1$^{ΔEC}$* mice that were administrated with r.Dll4 protein (Supplementary Fig. 5d). Our data demonstrate that Dll4/Notch1 signaling pathway is actively involved in the regulation of endothelial ZEB1 on bone angiogenesis and osteogenesis.

**Zeb1 gene delivery alleviates OVX-induced bone loss.** Evidence to date indicates that type H vessel numbers are significantly lower in the bone of osteoporotic humans and mice[4,6–8]. Notably, ZEB1 expression was significantly reduced in bone ECs of osteoporotic patients relative to normal controls, a finding accompanied by substantially decreased densities of total vessels and type H vessels (Fig. 7a, b). Based on these findings, we decided to determine whether pharmacological administration of exogenously expressed ZEB1 could ameliorate osteoporosis using a DNA-loaded cationic liposome vehicle[36–38] and an OVX mouse model[4,8].

To directly assess the efficiency and specificity of cationic liposome-delivered *Zeb1* gene in vivo, we generated a pcDNA3.1 +C-eGFP-ZEB1 vector (designed ZEB1-GFP) that can express recombinant enhanced greenfluorescent protein (eGFP)-fused ZEB1 protein in the target cells. Next, pcDNA3.1+C-eGFP (designed Vector-GFP) or ZEB1-GFP was mixed with cationic protamine at different N:P ratios (the molar ratio of nitrogen in protamine to phosphate in DNA vector) to form the nano-sized DNA/protamine complex (designed Protamine-Vector-GFP and Protamine-ZEB1-GFP). DNA packing efficiency was examined by agarose gel electrophoresis. The results showed that protamine can completely condense Vector-GFP and ZEB1-GFP at the N:P ratio of 3:1 and 4:1, respectively (Supplementary Fig. 6a). Protamine-Vector-GFP or Protamine-ZEB1-GFP was subsequently mixed with cationic liposomes at 60 nmol lipid/μg DNA to generate Lipo.-Vector-GFP and Lipo.-ZEB1-GFP, respectively. Agarose gel electrophoresis analysis revealed that Vector-GFP and ZEB1-GFP were completely encapsulated in the cationic liposomes under these conditions (Supplementary Fig. 6b). The final Lipo.-Vector-GFP had a particle size of 151.2 ± 5.7 nm (mean ± SD), a zeta potential of +24.9 ± 1.6 mv, and a polymer dispersity index of 0.286 ± 0.017, respectively, whereas the final Lipo.-ZEB1-GFP had 147.5 ± 4.9 nm, 23.8 ± 2.1 mv, and 0.270 ± 0.017, respectively (Supplementary Fig. 6c), confirming largely comparable physicochemical characteristics of Lipo.-Vector-GFP vs. Lipo.-ZEB1-GFP. We sought to analyze pharmacokinetic properties of the DNA-packaged liposomes. To this end, Vector-GFP and ZEB1-GFP were encapsulated into rhodamine phycoerythrin (PE)-labeled liposomes, and the DNA-packaged liposomes (designed Rho.-Lipo.-Vector-GFP and Rho.-Lipo.-ZEB1-GFP, respectively) were intravenously (i.v.) injected into rats. The blood samples collected at different times post

treatment were centrifuged and the supernatant plasma was collected for fluorescence intensity measurement. As shown, the fluorescence intensity–time curves of Rho.-Lipo.-Vector-GFP and Rho.-Lipo.-ZEB1-GFP were largely comparable (Supplementary Fig. 6d). Pharmacokinetic parameters such as $t_{1/2}$ (elimination half-life), $C_{max}$ (maximal plasma concentration), $AUC_{0-∞}$ (area under the plasma concentration–time curve), and MRT (mean residence time) of Rho.-Lipo.-Vector-GFP and Rho.-Lipo.-ZEB1-GFP were also comparable (Supplementary Fig. 6e). We next sought to analyze biodistribution of the DNA-packaged liposomes. For this purpose, ZEB1-GFP was packaged into 1,1'-dioctadecyl-3,3,3',3'-tetramethylindotricarbocyanine iodide (DiR)-labeled liposomes and the DNA-packaged liposomes (designed DiR-Lipo.-ZEB1-GFP) were i.v. injected into adult mice; the kidneys, spleen, lungs, brain, heart, liver, and hind limbs were dissected 24 h post injection and subjected to ex vivo biophotonic imaging analysis. As shown, the DiR signal was readily detected in the hind limbs, liver, spleen, lung, and kidneys, whereas the signal was hardly undetectable in the brain and heart (Supplementary Fig. 6f, g), validating efficient distribution of DiR-Lipo.-ZEB1-GFP to the skeletal and non-skeletal organs.

To generate OVX and sham-operation (Sham) mouse models, the surgery was performed on 8-week-old female mice and the mice at day 3 post surgery were i.v. injected with 4 μg Lipo.-Vector-GFP or Lipo.-ZEB1-GFP twice every week for consecutive 6 weeks. As predicted, uterine weight of OVX mice was reduced by 75% compared with Sham mice, whereas uterine weight of OVX mice treated with Lipo.-Vector-GFP or Lipo.-ZEB1-GFP was largely comparable (Supplementary Fig. 7a, b). To examine expressions of liposome-delivered Vector-GFP and ZEB1-GFP in target cells, the tibia and non-skeletal tissues were dissected for immunofluorescence analysis. The results revealed that liposome-delivered Vector-GFP and ZEB1-GFP were predominantly expressed in the metaphysis of the tibia (Fig. 7c) with remarkably lower expressions in the diaphysis of the tibia and non-skeletal tissues such as the lung, liver, and spleen (Supplementary Fig. 7c). Notably, a large proportion of GFP$^+$ cells was identified as EMCN$^+$ bone ECs, indicating that bone ECs in the metaphysis of the tibia are the major target cells of Lipo.-Vector-GFP and Lipo.-ZEB1-GFP (Fig. 7c). However, a small proportion of GFP$^+$ cells can be identified as EMCN$^-$ perivascular cells that may also respond to the liposome treatment (Fig. 7c). Furthermore, Vector-GFP recombinant protein was localized in cytoplasmic fraction of metaphyseal cells (in green; Fig. 7c, left and middle panels), whereas ZEB1-GFP recombinant protein was localized in the nucleus of metaphyseal cells (in yellow; Fig. 7c, right panels). In OVX mice treated with Lipo.-ZEB1-GFP, the positivity of endogenous ZEB1$^+$ bone ECs (i.e., the ratio of red/white: white) and exogenous ZEB1$^+$ bone ECs (i.e., the ratio of yellow/white:

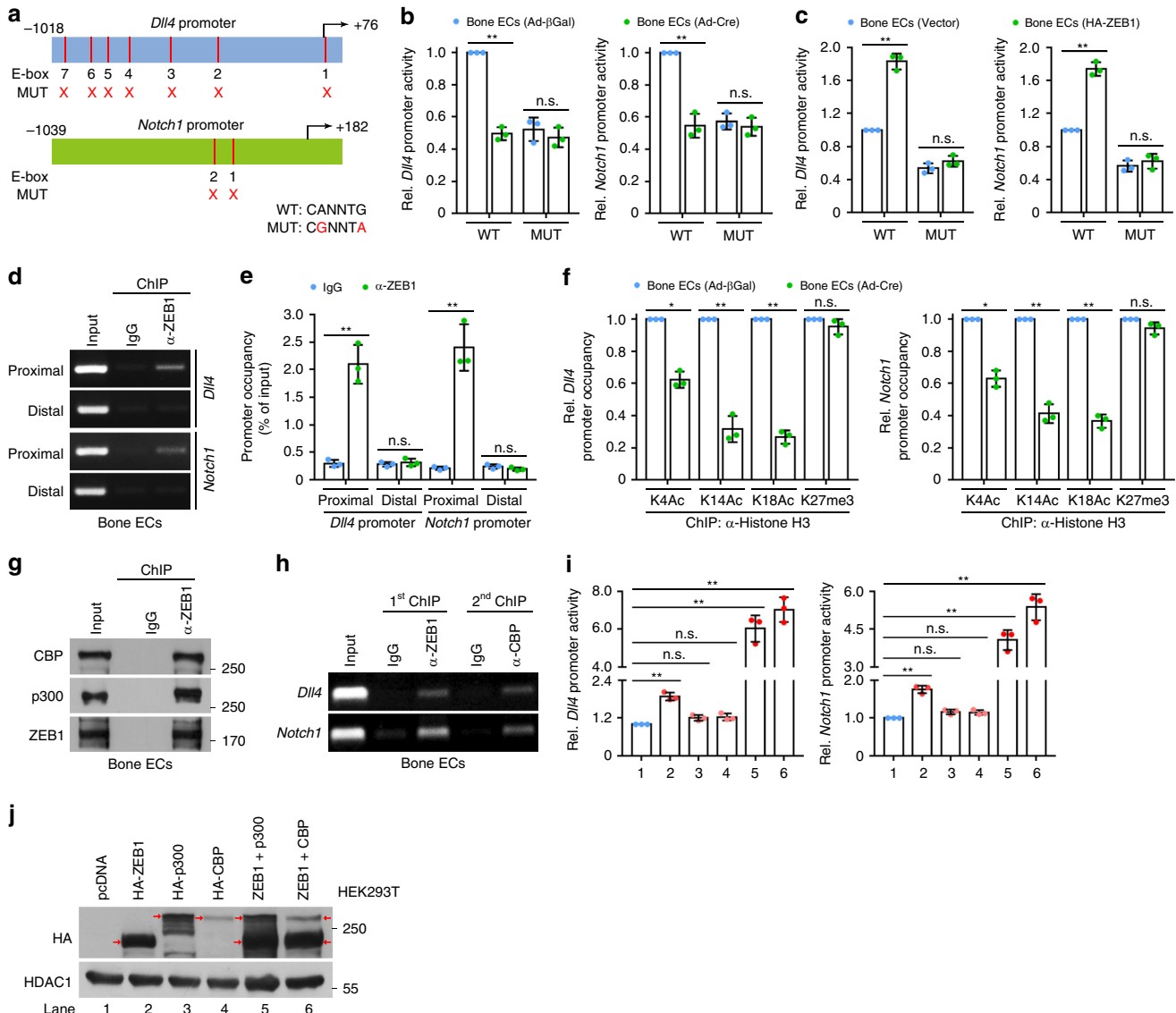

**Fig. 5 ZEB1 depletion epigenetically reduces *Dll4* and *Notch1* expressions in bone ECs. a** Schematic representation of reporter constructs of wild-type (WT) and mutated (MUT) murine *Dll4* and *Notch1* promoters. **b** Luciferase reporter analysis of cultured control and ZEB1-deleted bone ECs that were electroporated with WT or MUT *Dll4* and *Notch1* promoter reporter constructs ($n = 3$ independent experiments). **c** Luciferase reporter analysis of cultured bone ECs that were co-electroporated with WT or MUT *Dll4* and *Notch1* promoter reporter constructs in combination with pcDNA3.1 or pcDNA3.1-HA-ZEB1 ($n = 3$ independent experiments). **d**, **e** ChIP-PCR analysis of interactions of ZEB1-*Dll4* promoter and ZEB1-*Notch1* promoter within the proximal and distal regions in cultured bone ECs. Agarose gel images as shown are from three independent experiments (**d**) and promoter occupancy relative to corresponding input was quantified (**e**; $n = 3$ independent experiments). prox. proximal, dist. distal. **f** ChIP-qPCR for analyzing the enrichments of H3K4Ac, H3K14Ac, H3K18Ac, and H3K27me3 on *Dll4* and *Notch1* promoters in control and ZEB1-deleted bone ECs ($n = 3$ independent experiments). **g** Immunoprecipitation (IP) analysis of the physical interaction between ZEB1 and CBP or p300 in nuclear extracts of bone ECs. Images as shown are from three independent experiments. **h** Sequential ChIP-PCR analysis confirming the co-occupancy of ZEB1 and CBP or p300 on *Dll4* and *Notch1* promoters in bone ECs. Images as shown are from three independent experiments. **i**, **j** Luciferase reporter assays for analyzing *Dll4* and *Notch1* promoter activity in HEK293T cells that were co-transfected with the indicated constructs (**i**). Immunoblot analysis confirming ectopic expression of HA tagged ZEB1, p300, and CBP in nuclear extracts of HEK293T cells (**j**). Images as shown are from three independent experiments. Arrow marks specific bands with the expected molecular weights, respectively. All data are represented as mean ± SD. **$P < 0.01$, *$P < 0.05$; NS not significant. Differences are tested using one-way ANOVA with Tukey's post hoc test (**b**, **c**, **i**) and unpaired two-tailed Student's $t$-test (**e**, **f**). The source data are provided as a Source Data file. Unprocessed original scans of blots are shown in Source Data file.

white) was 13.4% and 28.3%, respectively, suggesting that exogenous recombinant ZEB1 protein was efficiently delivered to and expressed in EMCN$^+$ bone ECs (Fig. 7c, d). Endogenously expressed ZEB1 protein levels in bone ECs of OVX mice were decreased by 60% compared with Sham mice, and the levels of ZEB1 protein (including endogenously and exogenously expressed ZEB1) in bone ECs of OVX + Lipo.-ZEB1-GFP mice

were restored to 90% those of Sham + Vector-GFP mice (Fig. 7c, d). Importantly, the impaired type H vessel formation in tibial metaphysis of OVX mice was efficiently recovered following Lipo.-ZEB1-GFP treatment (Fig. 7e, f). Accordingly, the diminished expressions of Dll4 and Notch1 proteins in EMCN$^+$ bone ECs of OVX mice were also considerably restored following Lipo.-ZEB1-GFP treatment (Fig. 7g, h).

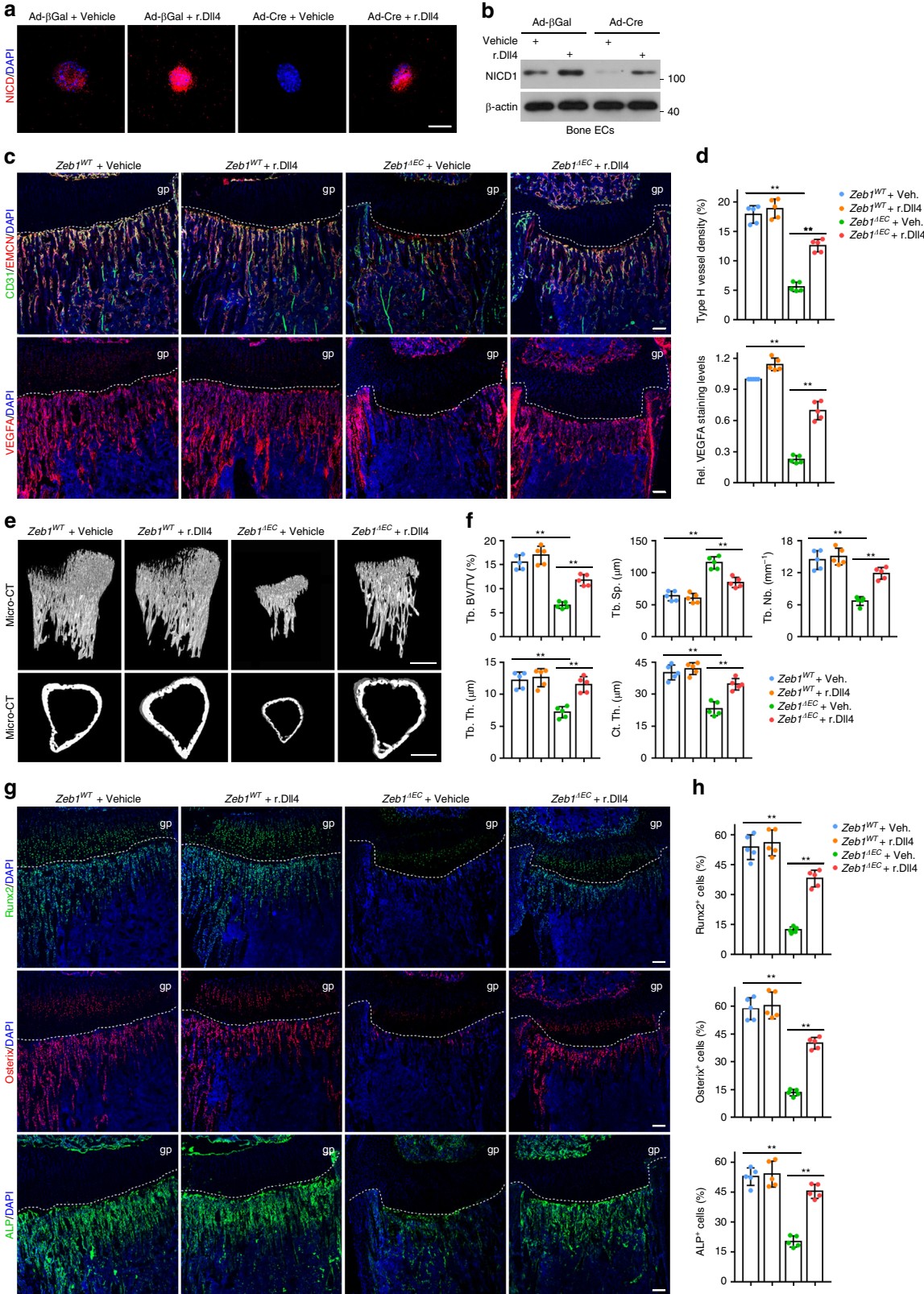

As predicted, micro-CT analysis demonstrated that tibia of OVX mice exhibited increased cross-sectional endosteal perimeters (might due to bone loss at the endosteum) in tandem with remarkably decreased Ct. Th., trabecular bone density, Tb. Nb., and Tb. Th. compared with Sham mice (Fig. 8a, b). Notably, administration of OVX mice with Lipo.-ZEB1-GFP efficiently restored the impaired bone formation in the mice (Fig. 8a, b). Immunofluorescence analysis further revealed that tibia of OVX mice had significantly lower numbers of Runx2[+] pre-osteoprogenitors and Osterix[+] osteoprogenitors than Sham mice (Fig. 8c, d), which is consistent with previous reports demonstrating that numbers of Runx2[+] and Osterix[+] cells are strongly

**Fig. 6 Administration of $Zeb1^{\Delta EC}$ mice with r.Dll4 protein efficiently restores the impaired bone angiogenesis and osteogenesis. a, b** Control and ZEB1-deleted bone ECs seeded on vehicle- or r.Dll4-precoated chamber wells (or culture dishes) were subjected to NICD1 immunostaining (**a**) and immunoblotting (**b**) analyses, respectively. Representative confocal or immunoblot images as shown are from three independent experiments. Scale bar, 20 μm. **c** Representative confocal images of CD31/EMCN and VEGFA immunostaining in the tibia of $Zeb1^{WT}$ and $Zeb1^{\Delta EC}$ mice that were i.p. injected with 1 μg/g r.Dll4 protein at P4 for 2 consecutive weeks before analysis at P21 ($n = 5$, each). Scale bar, 100 μm. **d** Quantification of type H vessel density (top panel) and VEGFA staining levels (bottom) in the tibia as shown in **c** ($n = 5$ independent experiments). **e** Micro-CT analysis of the tibia showing improved bone formation in trabecular bone (top panels) and cortical bone (bottom panels) of $Zeb1^{\Delta EC}$ mice treated with r.Dll4 protein as described in **c** ($n = 5$, each). Scale bar, 0.2 mm. **f** Quantification of bone architectures in the tibia as shown in **e** ($n = 5$ independent experiments). **g** Runx2, Osterix, and ALP immunostaining of the tibia as described in **c** ($n = 5$, each). Scale bar, 100 μm. **h** Quantification of Runx2[+], Osterix[+], and ALP[+] cells in the tibia as shown in **g** ($n = 5$ independent experiments). All data are represented as mean ± SD. **$P < 0.01$. Differences are tested using one-way ANOVA with Tukey's post hoc test (**d, f, h**). The source data are provided as a Source Data file. Unprocessed original scans of blots are shown in Source Data file.

reduced in OVX bone[39–41]. As predicted, administration of OVX mice with Lipo.-ZEB1-GFP efficiently normalized the numbers of Runx2[+] pre-osteoprogenitors and Osterix[+] osteoprogenitors (Fig. 8c, d). To evaluate whether repeated treatments with Lipo.-ZEB1-GFP or Lipo.-Vector-GFP could induce any histological alterations in major organs, we harvested major organs such as the lung, spleen, heart, liver, and kidney of liposome-treated mice for histo-morphometric analysis. Importantly, the results showed that administration of Sham or OVX mice with Lipo.-ZEB1-GFP or Lipo.-Vector-GFP for consecutive 6 weeks failed to induce any detectable histological alterations in the major organs, supporting the absence of Lipo.-ZEB1-GFP- or Lipo.-Vector-GFP-induced side toxicity (Supplementary Fig. 7d).

## Discussion

In the present study, we unexpectedly found that EMT-associated transcription factor (EMT-TF) ZEB1 is predominantly expressed in type H vascular endothelium in human and murine bone. Type H vessels primarily reside in the bone marrow near the growth plate but are essentially undetectable in non-skeletal organs[1–4]. Recent studies have demonstrated that type H endothelium impacts bone architecture, bone formation, and the recruitment of perivascular osteoprogenitor[1–4]. EC-specific deletion of ZEB1 strongly reduces type H vessel growth and bone formation but does not affect vasculature growth or cause histological alterations in the non-skeletal organs. The considerably variable consequences of ZEB1 depletion in ECs are due to the highly differential expression of ZEB1 in skeletal type H endothelium vs. non-skeletal endothelium. The EMT activator ZEB1 is largely a transcriptional repressor. ZEB1 silencing in mesenchymal cancer cells restores transcription of *CDH1* (encoding E-cadherin), a hallmark of EMT programs[10,11]. However, ZEB1 deletion in bone ECs does not affect *CDH1* transcription but indeed decreases histone acetylation on *Dll4* and *Notch1* promoters, thereby directly suppressing their transcriptional activity and down-regulating endothelial Notch signaling. The evolutionarily conserved Notch signaling pathway plays critical roles in controlling multiple aspects of vascular development and EC function, ranging from proliferation, motility, and lumen formation to vessel stability and cell fate determination[42,43]. Notch signaling has been identified as a critical positive regulator of vascular growth in bone, which is highly distinct from the Notch-mediated suppression of vessel sprouting in non-skeletal organs and in malignant tumors[44–46]. A recent report has demonstrated that ZEB1 indirectly upregulates Notch signaling by reducing miR-200 expression in multiple tumor cell lines[47]. Here we demonstrate that ZEB1 interacts with HATs CBP/p300 and they co-occupy the promoters of *Dll4* and *Notch1* to enhance histone acetylation on the promoters, thus directly inducing transcriptional activation of *Dll4* and *Notch1* in bone ECs. Our results further demonstrate that ZEB1/Notch signaling axis plays an essential role in controlling type H bone vessel formation. Notch signaling has been reported to promote bone-specific angiogenesis and osteogenesis

in both autocrine and paracrine manners[2]. Here we found that ZEB1/Notch signaling pathway controls recruitment/differentiation of perivascular osteoprogenitors and consequently promotes osteogenesis in the bone by regulating expression of multiple angiocrine factors such as TGFβ1, TGFβ2, BMP2, BMP4, FGF1, and Noggin. All these angiocrine factors are believed to actively participate in osteogenesis. Future work will address the mechanisms by which these angiocrine factors regulate differentiation and recruitment of osteoprogenitors in $Zeb1^{\Delta EC}$ bone microenvironment.

In the present study, we observe that the impaired bone angiogenesis and osteogenesis in OVX mice are efficiently restored following systematic treatment with ZEB1-GFP-packaged cationic liposomes. Although an efficient distribution of DNA-packaged cationic liposomes to the bone and various non-skeletal organs is observed, the liposome-delivered plasmid DNAs are predominantly expressed in the metaphysis of the tibia (mostly in type H bone ECs) with remarkably lower expressions in the diaphysis of tibia and non-skeletal elements. One possible explanation for the considerable difference between the organ distribution of DNA-packaged liposomes and expression of liposome-delivered plasmid DNA in target cells is that metaphyseal cells with highly metabolic activity[1] are more prone to uptake and express plasmids DNA than cells in the diaphysis of tibia and non-skeletal tissues. Consistent with this observation, no detectable histological alterations in non-skeletal organs such as the heart, liver, spleen, lung, or kidneys are observed after a 6-week course of systematic treatment of OVX mice with DNA-packaged cationic liposomes, supporting the absence of liposome-ZEB1-GFP- or liposome-Vector-GFP-induced toxicity or side effects. ZEB1 is inappropriately upregulated in a variety of malignant tumor types and functions as a stimulating factor of tumor invasion and metastasis[10–13]. Whether ectopic expression of ZEB1 in normal cells (e.g., metaphyseal cells) could induce transformation of the cells is still elusive. Future work will determine whether the extended period of treatment with $Zeb1$-packaged liposomes is still safe to the animals.

In conclusion, our results more broadly suggest that $Zeb1$-packaged cationic liposomes targeting bone vasculature may have a synergistic or complementary effect when used in combination with an osteoblast-targeted anabolic such as a parathormone analog or an anti-sclerostin antibody. The findings presented here lay the foundation for new therapeutic strategies in the treatment of osteoporotic disease states by promoting angiogenesis-dependent osteogenesis.

## Methods

**Mice.** Mice were housed under standard specific-pathogen-free conditions and all animal experiments were performed in accordance with protocols that were approved and authorized by the Animal Welfare and Ethics Committee of China Pharmaceutical University. Mice carrying *Zeb1* (exon 3) floxed alleles ($Zeb1^{fl/fl}$) were generated in our laboratory and have been described previously[15]. *Tie2-Cre* transgenic mice were purchased from Jackson Laboratory (#008863). *Cdh5(PAC)-CreERT2* transgenic mice were kindly provided by Ralf H. Adams (Max Planck

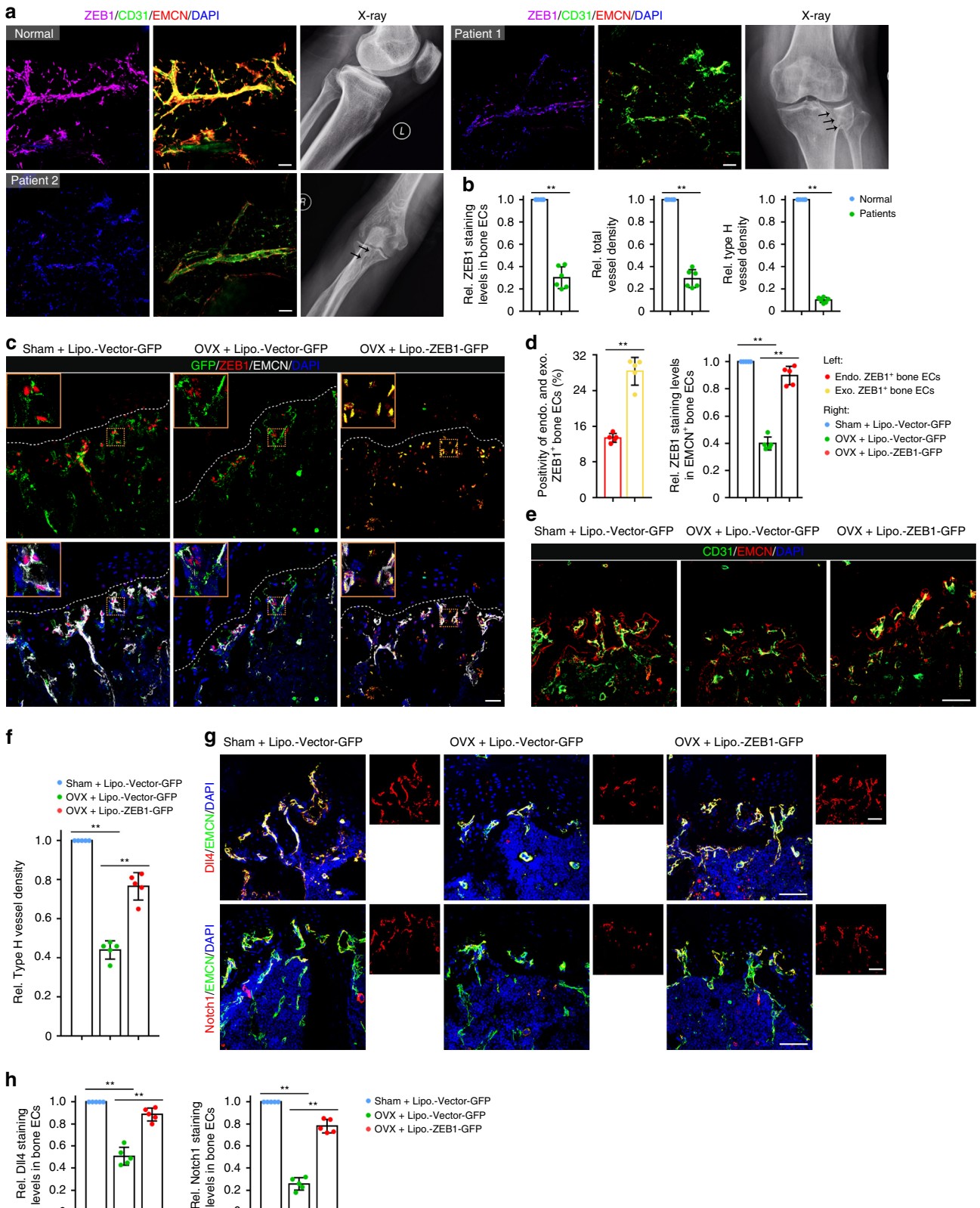

Institute for Molecular Biomedicine, Münster, Germany[18]). For generation of intrinsic and inducible EC-specific ZEB1-deleted mice, $Zeb1^{fl/fl}$ mice were bred with *Tie2-Cre* and *Cdh5(PAC)-CreERT2* transgenic mice, respectively. To induce Cdh5(PAC)-CreERT2 activity and gene inactivation in pups, mice at P8 were i.p. injected with 0.1 mg tamoxifen (10 μl, 10 mg/ml, dissolved in 1:10 ethanol/corn oil; Sigma-Aldrich, #T5648) every day for 7 consecutive days. To induce gene inactivation in adults, 7-week-old mice were i.p. injected with 1.0 mg tamoxifen (100 μl, 10 mg/ml) every other day for 2 consecutive weeks. For r.Dll4 administration

experiments, $Zeb1^{WT}$ and $Zeb1^{\Delta EC}$ mice were i.p. injected with 1 μg/g r.Dll4 protein (R&D Systems, #1389-D4-050) at P4 for 2 consecutive weeks before analysis at P21. To generate OVX and Sham mouse models, the surgery was performed on 8-week-old female mice, and the mice at day 3 post surgery were i.v. injected with 4 μg Lipo.-Vector-GFP or Lipo.-ZEB1-GFP twice every week for consecutive 6 weeks. Mice were then killed and uterine, long bone, and non-skeletal organs were dissected for further use. All mice were kept in C57BL/6J background and gender-matched littermate controls were used in all experiments. The investigators

**Fig. 7 Administration of OVX mice with *Zeb1*-packaged cationic liposomes efficiently restores the impaired bone angiogenesis. a** Representative confocal images of ZEB1/CD31/EMCN immunostaining in the tibia of normal and osteoporotic patients ($n = 6$, each). Arrow in X-ray image depicts tibial osteoporotic fracture undergoing knee joint replacement. Scale bar, 30 µm. **b** Quantification of ZEB1 staining levels in bone ECs, total vessel density, and type H vessel density in the tibia as shown in **a**. **c** Representative confocal images of GFP/ZEB1/EMCN immunostaining in the metaphysis of the tibia of Sham and OVX mice that were i.v. injected with 4 µg Lipo.-Vector-GFP or Lipo.-ZEB1-GFP for 6 consecutive weeks (designed Sham + Lipo.-Vector-GFP, OVX + Lipo.-Vector-GFP, and OVX + ZEB1-GFP mice, respectively; $n = 5$, each). Scale bar, 100 µm. **d** Quantification of positivity of endogenous and exogenous ZEB1$^+$ bone ECs (left panel) and ZEB1 staining levels in EMCN$^+$ bone ECs (right panel) in the tibia as shown in **c** ($n = 5$ independent experiments). **e** Representative confocal images of CD31/EMCN immunostaining in the tibia as shown in **c** ($n = 5$, each). Scale bar, 60 µm. **f** Quantification of type H vessel density in the tibia as shown in **c** ($n = 5$ independent experiments). **g** Representative confocal images of Dll4/EMCN (top panels) and Notch1/EMCN (bottom panels) immunostaining in tibia as shown in **c** ($n = 5$, each). Scale bar, 60 µm. **h** Quantification of Dll4 (left panel) and Notch1 (right panel) staining levels in bone ECs of the tibia as shown in **g** ($n = 5$ independent experiments). All data are represented as mean ± SD. **$P < 0.01$. Differences are tested using unpaired two-tailed Student's $t$-test (**b**) and one-way ANOVA with Tukey's post hoc test (**d**, **f**, **h**). The source data are provided as a Source Data file.

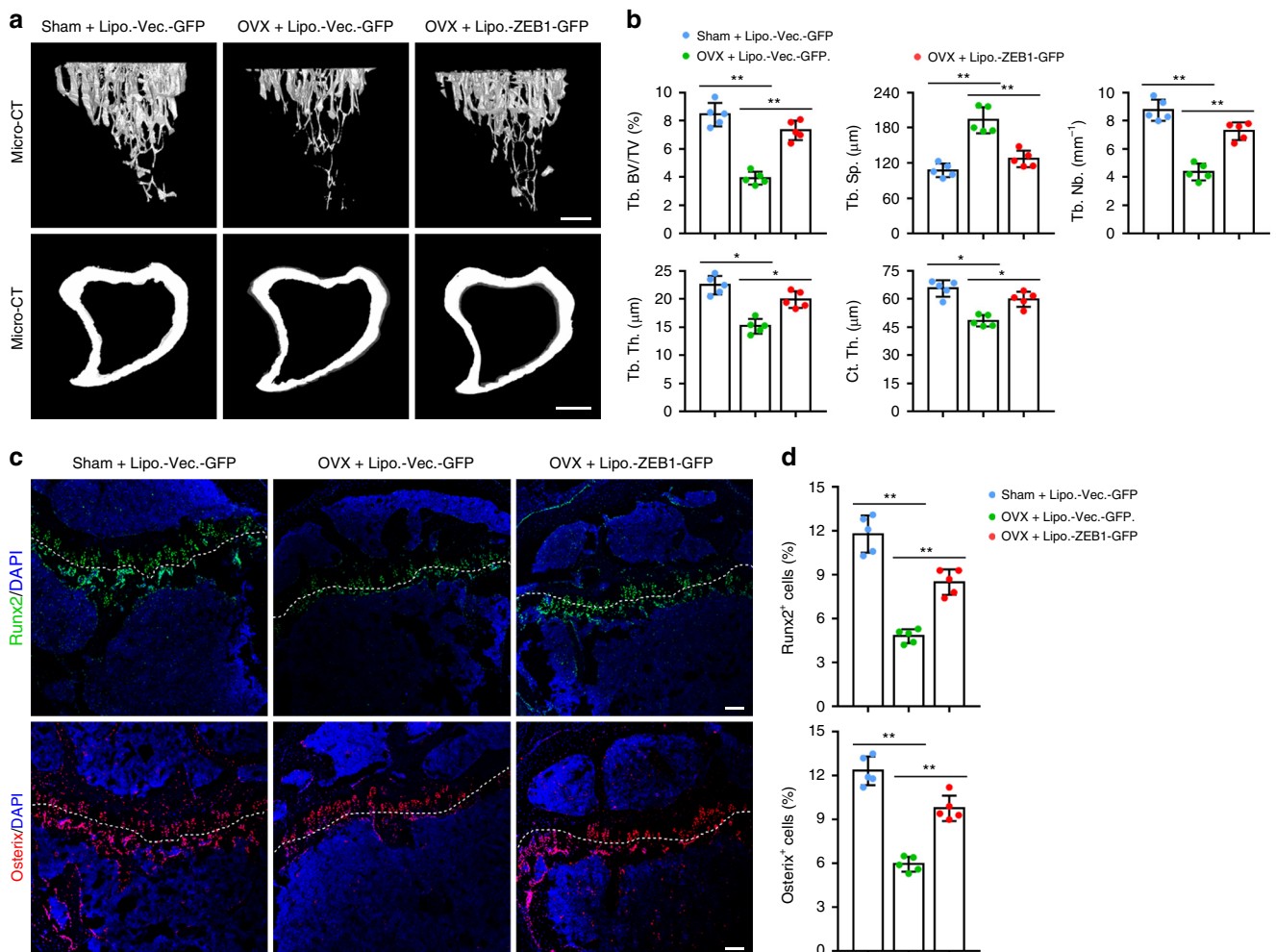

**Fig. 8 Treatment with *Zeb1*-packaged cationic liposomes has therapeutic effects on OVX-induced bone loss. a** Representative micro-CT images of trabecular bone (top panels) and cortical bone (bottom panels) of the tibia in Sham and OVX mice that were i.v. injected with 4 µg Lipo.-Vector-GFP or Lipo.-ZEB1-GFP for 6 consecutive weeks (designed Sham + Lipo.-Vector-GFP, OVX + Lipo.-Vector-GFP, and OVX + ZEB1-GFP mice, respectively; $n = 5$, each). Scale bar, 0.2 mm. **b** Quantification of bone architectures in the tibia as shown in **a** ($n = 5$ independent experiments). **c** Runx2 and Osterix immunostaining of the tibia as described in **a** ($n = 5$, each). Scale bar, 100 µm. **d** Quantification of Runx2$^+$ and Osterix$^+$ cells in the tibia as shown in **c** ($n = 5$ independent experiments). All data are represented as mean ± SD. **$P < 0.01$, *$P < 0.05$. Differences are tested using one-way ANOVA with Tukey's post hoc test (**b**, **d**). The source data are provided as a Source Data file.

were not blinded to allocation during experiments and outcome assessment. No method of randomization was used as mice were segregated into groups based on genotype alone. No statistical method was used to predetermine sample size.

**Genotyping**. Tail biopsy (~2 mm) samples were digested at 95 °C for 30 min in 50 µl of buffer 1 (10 N NaOH and 0.5 M EDTA pH 12.0). An equal amount of buffer 2 (40 mM Tris-HCl pH 5.0) was added to neutralize buffer 1. The mixtures were immediately vortexed and centrifuged at $12,000 \times g$ for 5 min. The supernatants (containing genomic DNAs) were collected and stored in −20 °C for further use. One microliter of extracted genomic DNA was added to 20 µl of PCR reaction mixture containing 10 µl 2× GoTaq Green Master Mix (Promega, #M7123), 0.5 µl forward/reverse primers, and 8.5 µl RNase/DNase free $H_2O$. Primer sequences for amplifying the indicated mouse transgenes are as follows.

*Zeb1-flox/wt* (flox-296 bp, wt-197 bp),
5′-GGTTTTACCATGCCCACTAATATGA-3′/5′-GAGGCAAGAAAAACAA ATGTAATCTCC-3′;

*Tie2-Cre* (PCR product, 450 bp),
5′-AGGTGTAGAGAAGGCACTTAGC-3′/5′-CTAATCGCCATCTTCCAGGA GG-3′;

*Cdh5(PAC)-CreERT2* (PCR product, 720 bp),
5′-GCCTGCATTACCGGTCGATGCAACGA-3′/5′-GTGGCAGATGGCGCG GCAACACCATT-3′.

**Immunofluorescence and histology**. For immunofluorescence, tissues were freshly dissected and fixed in 4% paraformaldehyde (PFA)–phosphate-buffered saline (PBS) at 4 °C for 4 h with constant shaking. Bone samples were decalcified in 0.5 M EDTA at 4 °C and the decalcified bones were immersed into 20% sucrose and 2% polyvinylpyrrolidone solution for 24 h. Bone and non-skeletal tissues were embedded in OCT (Sakura, #4583) and 50 μm-thick sections were prepared. Paraffin-embedded sections were deparaffinized and treated with the Antigen Unmasking Solution (#H-3300, Vector). Tissue sections were permeabilized for 15 min in 0.3% Triton X-100. After washing, sections were blocked with 5% donkey serum (in PBS) at room temperature for 30 min and incubated overnight at 4 °C with primary antibodies. Cryostat sections were air-dried, permeabilized in 0.3% Triton X-100 for 15 min, blocked in 5% goat or donkey serum at room temperature for 30 min, and probed with primary antibodies diluted in 5% goat or donkey serum in PBS overnight at 4 °C. The following antibodies were used: ZEB1 (Santa Cruz, #sc-25388, 1:200), Endomucin (Santa Cruz, #sc-65495, 1:200), CD31 conjugated to Alexa Fluor 488 (R&D Systems, #FAB3628G, 1:100), CD31 (BD Pharmingen, #553370, 1:200), Osterix (Santa Cruz, #sc-22536, 1:200), Runx2 (Abcam, #ab192256, 1:200), VEGFA (Abcam, #ab52917, 1:200), Notch1 (Cell Signaling, #4380, 1:100), Dll4 (R&D Systems, #AF1389, 1:100), NICD1 (Abcam, #ab8925, 1:100), ALP (R&D Systems, #MAB29091, 1:200), and GFP (Abcam, #ab5450, 1:200). After incubation with primary antibodies, sections were washed with PBS three times and were incubated with species-appropriate Alexa Fluor-coupled secondary antibodies (all from Abcam, 1:200) for 1 h at room temperature. Sections were then counterstained with 4′,6-diamidino-2-phenylindole (DAPI) and images were acquired on a Zeiss LSM800 microscope.

For calcein double labeling, mice were i.p. injected with 10 mg/kg calcein (Sigma-Aldrich, #C0875) dissolved in 2% sodium bicarbonate solution at P13 and P19, and bone samples were collected at P21, fixed in 4% PFA, directly embedded in methyl methacrylate, and sagittally sectioned at 30 μm. Images of calcein double labeling in undecalcified tibial sections were acquired on a Zeiss LSM800 confocal microscope.

For histology, non-skeletal tissues were fixed in 4% PFA–PBS and embedded in paraffin. Bone samples were fixed in 4% PFA–PBS, decalcified as described above, and embedded in paraffin. The embedded tissues were sectioned at 5 μm, deparaffinized, and subjected to hematoxylin and eosin staining and tartrate-resistant acid phosphatase (TRAP) staining using an acid phosphatase leukocyte kit (Sigma-Aldrich, #386 A) according to the manufacturer's instruction. In some experiments, bone samples were fixed in 4% PFA–PBS, directly embedded in paraffin, sectioned, and subjected to Alizarin red S staining (Sigma-Aldrich, #A5533).

**Micro-CT analysis**. The tibias were freshly dissected, fixed in 4% PFA–PBS, and analyzed using high-resolution micro-CT (Hiscan Information Technology, #Hiscan-H100) and software Hiscan Analyzer V3.0. The scanner was set at a voltage of 70 kV, a current of 110 mA, and a resolution of 10.8 μm/pixel. The metaphysis of the tibia was scanned and metaphyseal parameters such as Tb. BV/TV, Tb. Th., trabecular separation (Tb. Sp.), Tb. Nb., and cortical bone thickness (Ct. Th.) were measured.

**Isolation and culture of bone ECs**. To isolate primary bone ECs, the tibias were collected in sterile $Ca^{2+}$ and $Mg^{2+}$ free PBS, crushed, and digested in collagenase A (Sigma-Aldrich, #SCR136) for 30 min at 37 °C to obtain a single-cell suspension. Bone ECs were then magnetic activated cell sorting sorted using Endomucin antibody (Santa Cruz, #sc-65495) and Dynabeads sheep anti-Rat IgG (Thermo Fisher, #11035). Sorted bone ECs were seeded on fibronectin-precoated culture dishes and cultured in EBM-2 EC growth medium (Lonza, #00190860) containing supplements and growth factors (Lonza, #CC-4176). Normally, bone ECs were used within five passages. In some experiments, bone ECs isolated from the tibia of ZEB1fl/fl mice were infected with recombinant adenovirus expressing βGal (Vigene Biosciences, #CV1001) or Cre (#CV10010) at a multiplicity of infection of 200 for 24 h and re-infected with for additional 24 h to generate control or ZEB1-deleted cells.

**Flow cytometry**. Tibias were collected in sterile $Ca^{2+}$ and $Mg^{2+}$ free PBS, crushed, and digested in collagenase A (Sigma-Aldrich, #SCR136) for 30 min at 37 °C to obtain a single-cell suspension. Equal number of cells were immunostained with Endomucin antibody (Santa Cruz, #sc-65495) for 45 min at 4 °C. After washing, cells were immunostained with PE-CD45 (BD Pharmingen, #553081), APC-Ter119 (BD Pharmingen, #557909), PE-Cy7-CD31 (BD Pharmingen, #561410), and Alexa Fluor 647 goat anti-rat secondary antibody (Thermo Fisher, #A-21247) for 45 min

at 4 °C, followed by DAPI staining for 5 min before analysis and sorting. Finally, cells were acquired on a BD FACS Verse flow cytometer and sorted on a FACS Aria II flow cytometer. To demarcate and sort CD31hiEMCNhi cells, first standard quadrant gates were set. To differentiate CD31hiEMCNhi cells from the total double positive cells in quadrant, two gates were set at >$10^4$ log Fl-2 (PE-Cy7-CD31) fluorescence and >$10^4$ log Fl-4 (Alexa Fluor 647-Endomucin) fluorescence. DAPI−CD45−Ter119−CD31+ cells were sorted according to side scatter and were set at >$10^3$ log Fl-2 fluorescence (PE-Cy7-CD31) after negative selection of CD45 and Ter119 at <$10^3$ log Fl-2 (PE-CD45 and APC-Ter119) fluorescence. DAPI−CD45−Ter119−CD31+ cells were sorted as total bone ECs.

**Quantitative reverse-transcriptase PCR**. Total RNA was isolated and reversely transcribed using the RNeasy kit (Qiagen, #74104) and the PrimeScript RT reagent kit (TaKaRa, #RR037A), respectively, according to the manufacturer's instructions. qPCR was performed using the SYBR Green PCR Master Mix (Vazyme, #Q341) and relative mRNA expression was normalized to *Gapdh*. Primer sequences are as follows.

*Zeb1*, 5′-TTATCCTGAGGCGCCCGAGGA-3′/5′-TACGGGCAGGTGAGCAA CTGG-3′;

*Zeb2*, 5′-GCACCCAGCTCGAGAGGCAT-3′/5′-AAGGCCTTGCCACACTCC GTG-3′;

*Notch1*, 5′-TGCCTGAATGGAGGTAGGTGC-3′/5′-GCACAGCGATAGGAG CCGATC-3′;

*Notch2*, 5′-TGACTGTTCCCTCACTATGG-3′/5′-CACGTCTTGCTATTCCTC T-3′;

*Notch3*, 5′-TGCCAGAGTTCAGTGGTGG-3′/5′-TGCCAGAGTTCAGTGGT GG-3′;

*Notch4*, 5′-CTCTTGCCACTCAATTTCCCT-3′/5′-TTGCAGAGTTGGGTATC CCTG-3′;

*Dll4*, 5′-GACTGAGCTACTCTTACCGGG-3′/5′-CTTACAGCTGCCACCATT TCG-3′;

*Hes1*, 5′-GGAGAAGAGGCGAAGGGCAAG-3′/5′-GGTTCCGGAGGTGCTT CACAG-3′;

*Hes5*, 5′-GAAACACAGCAAAGCCTTCG-3′/5′-AGCTTCATCTGCGTGTC G-3′;

*Hey1*, 5′-GGCTGGTACCCAGTGCCTTTG-3′/5′-CCTTTCCCTCCTGCAGT GTGC-3′;

*Heyl*, 5′-CAGCCCTTCGCAGATGCAA-3′/5′-CCAATCGTCGCAATTCAGA AAG-3′;

*Efnb2*, 5′-TGGGTCTTTGGAGGGCCTGGAT-3′/5′-GGACCGTGATTCCTG GCTGATC-3′;

*Tgfb1*, 5′-CTCCCGTGGCTTCTAGTGC-3′/5′-GCCTTAGTTTGGACAGGAT CTG-3′;

*Tgfb2*, 5′-TCGACATGGATCAGTTTATGCG-3′/5′-CCCTGGTACTGTTGTA GATGGA-3′;

*Tgfb3*, 5′-CAGGCCAGGGTAGTCAGAG-3′/5′-ATTTCCAGCCTAGATCCTG CC-3′;

*Bmp2*, 5′-GGGACCCGCTGTCTTCTAGT-3′/5′-TCAACTCAAATTCGCTGA GGAC-3′;

*Bmp4*, 5′-TTCCTGGTAACCGAATGCTGA-3′/5′-CCTGAATCTCGGCGACT TTTT-3′;

*Pgf*, 5′-TCTGCTGGGAACAACTCAACA-3′/5′-GTGAGACACCTCATCAGG GTAT-3′;

*Fgf1*, 5′-CCCTGACCGAGAGGTTCAAC-3′/5′-GTCCCTTGTCCCATCCAC G-3′;

*Vegfa*, 5′-GCACATAGAGAGAATGAGCTTCC-3′/5′-CTCCGCTCTGAACAA GGCT-3′;

*Nog*, 5′-GCCAGCACTATCTACACATCC-3′/5′-GCGTCTCGTTCAGATCCT TCTC-3′;

*Gapdh*, 5′-CCCTGGCCAAGGTCATCCATG-3′/5′-TGATGTTCTGGGCAGC CCCAC-3′.

**Immunoblot and immunoprecipitation assays**. Immunoblot assay was performed as described previously[15]. Briefly, cells were collected and lysed in RIPA buffer (Thermo Fisher, #89900) containing protease inhibitor cocktail (Thermo Fisher, #87786). In some experiments, nuclear proteins were extracted using the NE-PER Nuclear and Cytoplasmic Extraction Reagents (Thermo Fisher, #78833) according to the manufacturer's instructions. The cell lysates were subjected to immunoblot assay using primary antibodies against ZEB1 (Santa Cruz, #sc-25388, 1:1,000), Dll4 (R&D Systems, #AF1389, 1:1,000), Notch1(Cell Signaling, #4380, 1:1,000), NICD1 (Cell Signaling, #4147, 1:500), CBP (Cell Signaling, #7389, 1:1,000), p300 (Cell Signaling, #70088, 1:1,000), HA (Cell Signaling, #3724, 1:1,000), HDAC1 (Cell Signaling, #2062, 1:1,000), and β-actin (Santa Cruz, #sc-47778, 1:2,000). IP assay was performed as described previously[15]. In brief, cells were collected and lysed in IP lysis buffer (50 mM Tris-HCl, 150 mM NaCl, 1 mM EDTA, 1% NP40, pH 7.4) supplemented with protease inhibitor cocktail for 20 min on ice. The cell lysates were sonicated, clarified, and incubated with anti-ZEB1 antibody (Santa Cruz, #sc-25388, 1:200) followed by incubation with pre-cleared Protein A/G agarose beads (Santa Cruz, #sc-2003). The immunocomplexes were

subjected to immunoblotting using antibodies against CBP, p300, and ZEB1. The original uncropped scans of western blottings are shown in Source Data file.

**ChIP and sequential ChIP assays.** Bone ECs were fixed in 1% formaldehyde for 10 min, quenched in 0.125 M glycine for 5 min at room temperature, washed with ice-cold PBS for three times, and lysed in lysis buffer (50 mM Tris-HCl, 150 mM NaCl, 1 mM EDTA, 1% NP40 pH 7.4) supplemented with protease inhibitor cocktail for 20 min on ice. The cell lysates were sonicated until the sheared DNA was ~200–1000 bp in size. After centrifugation, the supernatants were pre-cleared with protein A/G beads and saturated with salmon sperm DNA (Thermo Fisher, #15632011) at 4 °C for 1 h. Five percent of the sheared chromatin was used as input control and the rest was incubated overnight with the antibodies against ZEB1 (Santa Cruz, #sc-25388), acetyl-histone H3 (K4; Cell Signaling, #7627), acetyl-histone H3 (K18; Cell Signaling, #9675), acetyl-histone H3 (K14; Abcam, #ab176799), and tri-methyl-histone H3 (K27; Cell Signaling, #9377) at 4 °C under rotation, followed by another incubation with protein A/G beads (Santa Cruz, #sc-2003) for 1 h at 4 °C. Beads-bound chromatins were sequentially washed with low-salt wash buffer once, high-salt wash buffer once, and TE buffer twice, and were eluted with elution buffer (1% SDS, 0.1 M NaHCO$_3$) twice. The eluted DNA-protein complexes were incubated with 0.2 M NaCl overnight at 65 °C, RNase A for 30 min at 37 °C, and proteinase K (Thermo Fisher, #AM2548) for 1.5 h at 45 °C. The bound DNA was purified using a kit (Sigma-Aldrich, #NA1020) according to the manufacturer's instructions and then subjected to PCR or qPCR analysis. For sequential ChIP assay, following sequential washes, half amount of beads were eluted as described above and saved for further use. Another half amount of beads were eluted with 2% dithiothreitol buffer at 37 °C for 30 min. The supernatants were collected and incubated with anti-CBP antibody (Cell Signaling, #7389) or IgG isotype control (Santa Cruz, #sc-2763) overnight at 4 °C, followed by another incubation with protein A/G beads for 1 h at 4 °C. Beads-bound chromatins were sequentially washed with low-salt wash buffer once, high-salt wash buffer once and TE buffer twice and eluted with elution buffer (1% SDS, 0.1 M NaHCO$_3$) twice. The eluted DNA-protein complexes and the resulting bound DNA were treated as described above. Primer sequences are as follows.

*Dll4*-proximal (PCR product, 81 bp),
5′-CAGCCTCAAGCTCTCTCACC-3′/5′-CGCTGGAGTAGGGAGGAAAC T-3′.

*Dll4*-distal (PCR product, 90 bp),
5′-TTGCCTAGAGGGAAAGAAAGG-3′/5′-GCGATCTGGGAGACTGTATT G-3′.

*Notch1*-proximal (PCR product, 104 bp),
5′-GCCAGGGCGCCACATTTAAAC-3′/5′-GAACCAGCTCCATCCTGGA GT-3′.

*Notch1*-distal (PCR product, 85 bp),
5′-AAGGGGAGCGAGTGAATGAGG-3′/5′-GCTCAGTACCAGCTCAGCC TT-3′.

**Luciferase reporter assay.** The ~1.1 kb fragment of WT or MUT mouse *Dll4* promoter (−1018 ~ +76) and the ~1.2 kb fragment of WT or MUT mouse *Notch1* promoter (−1039 ~ +182) were synthesized by GenScript and subcloned into the pGL3 basic luciferase reporter vector (Promega, #E1751). Control or ZEB1-deleted bone ECs were eletroporated with *Dll4*-WT (or *Dll4*-MUT, *Notch1*-WT, or *Notch1*-MUT) reporter constructs and pRL-TK *Renilla* luciferase control construct. Bone ECs were also eletroporated with pcDNA3.1 (or pcDNA3.1-HA-ZEB1), *Dll4*-WT (or *Dll4*-MUT, *Notch1*-WT, or *Notch1*-MUT) reporter constructs, and pRL-TK *Renilla* luciferase control construct. HEK293T cells were transfected with pcDNA3.1, pcDNA3.1-HA-ZEB1, pSG5-HA-p300 (Addgene, #89094), pRC/RSV-mCBP-HA (Addgene, #16701), ZEB1 + p300, ZEB1 + CBP together with *Dll4*-WT or *Notch1*-WT promoter reporter constructs, and pRL-TK *Renilla* luciferase control construct. Dual-luciferase reporter assays were performed according to the manufacturer's instructions (Promega, #E1910).

**r.Dll4 treatment.** Culture dishes were precoated with 500 ng/ml r.Dll4 protein (R&D Systems, #1389) dissolved in PBS overnight at 4 °C and control and ZEB1-deleted bone ECs were seeded on r.Dll4-precoated culture dishes and grown for 24 h. Cells were collected and total RNAs were isolated for RT-qPCR analysis.

**Human bone samples.** Human bone samples were obtained from patients (female) based on the inclusion and exclusion criteria. Six patients with osteoporosis undergoing knee joint replacement with ages ranging from 58 to 75 years and six patients with tibia fracture undergoing Open Reduction Internal Fixation ranging in age from 12 to 30 years (human bone samples were collected at the Division of Orthopedic Surgery, the Affiliated Nanjing Hospital, Nanjing Medical University, Nanjing, China). This clinical study was approved by the Ethnic Committee of the Affiliated Nanjing Hospital, Nanjing Medical University, and written informed consents were obtained from the patients before procedure.

**DNA-packaged liposome preparation and characterization.** Full length of murine *Zeb1* cDNA were synthesized by GenScript and subcloned into pcDNA3.1 +C-eGFP (designed Vector-GFP) vector to generate pcDNA3.1+C-eGFP-ZEB1

(designed ZEB1-GFP). For preparation of cationic liposomes, 1,2-dioleoyl-*sn*-glycero-3-phosphoethanolamine, 1,2-dioleoyl-3-trimethylammonium propane, and cholesterol were dissolved in chloroform at a molar ratio of 4:3:3, evaporated in a rotary evaporator at 40 °C for 30 min, dried overnight, hydrated with dd H$_2$O at 40 °C for 20 min, and finally ultrasounded at an intensity of 15%. Vector-GFP (0.2 mg/ml) or ZEB1-GFP (0.2 mg/ml) was mixed with cationic protamine at different N:P ratios (the molar ratio of nitrogen in protamine to phosphate in DNA vector) to form the nano-sized DNA/protamine complex (designed Protamine-Vector-GFP and Protamine-ZEB1-GFP). The Protamine-Vector-GFP and Protamine-ZEB1-GFP was subsequently mixed with cationic liposomes at 60 nmol lipid/μg DNA to generate the final Lipo.-Vector-GFP and Lipo.-ZEB1-GFP, respectively. DNA packing efficiency was examined by agarose gel electrophoresis. Particle size and zeta potential were measured using a Zetasizer NANO ZSP (Malvern Panalytical). For preparation of fluorescence-loaded liposomes, *N*-(lissamine rhodamine B sulfonyl)-1,2-dihexadecanoyl-sn-glycero-3-phosphoethanolamine (2 mol%) and DiR (0.1 mol%) were added in the lipid components.

**Pharmacokinetics.** For pharmacokinetic analysis, pcDNA3.1+C-eGFP control vector or pcDNA3.1+C-eGFP-ZEB1 vector was packaged into rhodamine PE-labeled liposome and the DNA-packaged liposomes (designed Rho.-Lipo.-Vector-GFP and Rho.-Lipo.-ZEB1-GFP, respectively) were i.v. injected into Sprague–Dawley rats ($n = 5$, each) at a dose of 0.8 mg/kg; The blood samples collected at 0, 0.125, 0.25, 0.5, 1, 2, 4, 8, 12, and 24 h post treatment, were centrifuged, and the supernatant plasma was collected for fluorescence intensity measurement at the excitation wavelength of 552 nm and the emission wavelength of 610 nm on a SpectraMax iD5 Multi-Mode Microplate Reader (Molecular Devices). The fluorescence intensity–time curves of Rho.-Lipo.-Vector-GFP and Rho.-Lipo.-ZEB1-GFP were plotted. The pharmacokinetic parameters such as elimination half-life ($t_{1/2}$, h), maximal plasma concentration ($C_{max}$, × 10$^4$ a.u./ml), area under the plasma concentration–time curve (AUC$_{0-\infty}$, × 10$^4$ a.u./ml × h), and MRT (h) were calculated using Phoenix WinNonlin 6.4 software.

**Biodistribution measurement.** For biodistribution analysis, pcDNA3.1 + C-eGFP-ZEB1 vector was packaged into DiR-labelled liposome and the DNA-packaged liposomes (designed DiR-Lipo.-ZEB1-GFP) were i.v. injected into 8-week-old C57BL6 mice at a dose of 1.0 mg/kg; the kidneys, spleen, lungs, brain, heart, liver, and hind limbs were dissected 24 h post injection and were subjected to biophotonic imaging assay for evaluation of organ distribution of DiR-Lipo.-ZEB1-GFP using the IVIS Spectrum In Vivo Imaging System (PekinElmer). The region of interest was circled in the collected tissues and the fluorescence intensity of DiR was measured using Living Image software.

**Statistics.** Data were presented as mean ± SD. Statistical analysis was carried out as described in each corresponding figure legend and sample sizes were shown in each corresponding figure legend. $P < 0.05$ is considered significant.

**Reporting summary.** Further information on research design is available in the Nature Research Reporting Summary linked to this article.

## Data availability

The source data underlying Figs. 1b, 1d, 1e, 1g, 1i, 2c, 2d, 2e, 2g, 2h, 2k, 2n, 2p, 2r, 2t, 2v, 3a, 3d, 3e, 3g, 4b, 4c, 4e, 4f, 4i, 5d, 5f, 5h, 6b, 6d, 6f, 6h, 7b, and 7d, and Supplementary Figs. 1c, 1e, 1g, 2b, 2d, 2f, 3a, 3c, 3f, 3h, 3j, 4b, 4d, 4f, 4h, 5a, 5c, 5d, 6b, 6c, 6d, 6f, and 7b are provided as a Source Data file. Unprocessed original scans of blots are shown in Supplementary Fig. 8. The remaining data are contained within the Article, Supplementary Information, or available from the authors upon request. A reporting summary for this Article is available as a Supplementary Information file.

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

## Acknowledgements

This research was supported by grants from Natural Science Foundation of China (81973363, 81572745, 91539115, 81603134, 81771985, 81702205, and 81702163), the Jiangsu Provincial Natural Science Funds for Distinguished Young Scholar (BK20170029), the Jiangsu Provincial Natural Science Funds for Young Scholar (BK20160758), the Jiangsu Provincial Innovative Research Program, the State Key Laboratory of Natural Medicines of China Pharmaceutical University (SKLNMZZCX201808), and the "Double First-Class" University project (CPU2018GF02). We thank Dr R. Adams (Max Planck Institute for Molecular Biomedicine, Germany) for providing Cdh5(PAC)-CreERT2 mice and Dr M.-D. Lai (China Pharmaceutical University, China) for critically reviewing this manuscript.

## Author contributions

Z.-Q.W. conceived the project, designed experiments, interpreted results, and wrote the manuscript. R.F. designed and performed experiments, interpreted results, and wrote the manuscript. W.-C.L., Y.X. and M.-Y.G. performed experiments with the help from X.-J.C., N.J., Y.X., L.D. and Q.-Q.Y. T.L., L.-M.W. and R.M. provided relevant advice.

## Competing interests

Z.-Q.W., R.F. and R.M. are inventors on a pending patent related to this work. The other authors declare no competing interests.
