## [Peer Review File · Nature Communications]

Reviewers' Comments:

Reviewer #1:

Remarks to the Author:

In this manuscript, the authors used mice with endothelial cell-specific deletion of Zeb1 to demonstrate that Zeb1 plays an important role in CD31-high, endomucin-high vessel formation in the bone and osteogenesis. Mechanistically, Zeb1 deletion reduces histone acetylation on Dll4 and Notch1 promoters, leading to decreased Notch signaling, a critical pathway in angiogenesis and osteogenesis. In addition, add-back of Zeb1 by Zeb1 liposome reversed Zeb's loss-of-function effects on Notch signaling and angiogenesis-dependent osteogenesis. Also, administration of recombinant Dll4 protein also reversed Zeb1 deletion-induced phenotypes (restored CD31-high EMCN-high angiogenesis and promoted bone formation) in Zeb1 conditional knockout mice. Overall, this is a rigorous study demonstrating the functional importance of Zeb1 in bone angiogenesis and osteogenesis. Here are my comments on the mechanism of Zeb1 in angiogenesis:

1. Figure 3a: it is not clear why the authors specifically focused on the Notch pathway. The Zeb1/miR-200 signaling axis was reported to regulate Notch signaling (EMBO J. 2011 Feb 16; 30(4): 770–782). Does this axis contribute to the role of Zeb1 in angiogenesis?
2. Why does Zeb1 deletion specifically affect angiogenesis in the bone but not in other tissues?
3. In addition to Notch signaling, does Zeb1 deletion affect other pathways in CD31-high, endomucin-high endothelial cells? Unbiased profiling (RNA-seq) and pathway analysis should be performed.
4. Figure 3p and 3q: does knockout of Zeb1 specifically regulate histone acetylation on Dll4 and Notch1 promoters? How about other regions of the genome? Unbiased analysis (ChIP-seq) should be performed.
5. What is the mechanism by which Zeb1 regulates histone acetylation?

Reviewer #2:

Remarks to the Author:

In their manuscript "Endothelial Zeb1 promotes angiogenesis-dependent bone formation and reverses osteoporosis", Fu et al. report that the zinc-finger transcription factor Zeb1 controls Notch signalling in bone endothelial cells (ECs) and is thereby critical for the coupling of angiogenesis and osteogenesis. The authors have generated and characterized EC-specific Zeb1 mutant mice, and their data convincingly shows that both vessel growth and osteogenesis in long bone are impaired relative to littermate controls. At the molecular level, loss of Zeb1 leads to reduced expression of Notch pathway genes and other data support that this is due to binding of Zeb1 to various E-box motifs in the Dll4 and Notch1 promoters. The authors also show that treatment of Zeb1 mutant mice with a soluble, recombinant version of Dll4 restores Notch signalling and improves osteogenesis. A final set of experiments indicates that Zeb1 expression is reduced in osteoporotic patients and in the ovariectomy mouse model. Administration of Zeb1-packaged liposomes increases both the number of osteoprogenitor cells and the amount mineralized bone in ovariectomized mice, suggesting that Zeb1 might be a promising therapeutic target.

Previous work by other has already shown that Zeb1 is an important regulator of craniofacial and skeletal development, but it was unclear whether the protein plays a role in the endothelium or in other cell types in bone. There are also reports about positive regulation of VEGF-A expression and Notch signalling by Zeb1, which are obviously relevant for this study and should be at least mentioned in the Discussion.

Overall, the data is mostly of good quality and it is easy to follow the logic of the manuscript. There are, however, a couple of important issues that need to be addressed prior to publication:

I could not find the protocol for the tamoxifen administration anywhere in the manuscript. The wording of the third paragraph in the Online Methods is rather confusing. Presumably, mutants generated with the Tie2-Cre line were not treated with tamoxifen. It is also not stated whether control mice in the experiments with the Cdh5-CreERT2 line were also treated with tamoxifen. These issues should be clarified.

I am concerned about the specificity of the Dll4 immunostaining in Fig. 3b. Instead of the expected EC-specific staining, a lot of signal can be seen in perivascular cell populations. This, the alterations in the mutant samples might simply reflect a reduction of perivascular cells but not necessarily impaired Dll4 expression. This is a critical issue with relevance for the main conclusions of the study.

It is noteworthy that the cellularity of the bone marrow is strongly reduced in EC-specific Zeb1 mutants (see Suppl. Fig. 4b; bottom panels). What is the cause for this defect? Is there a higher abundance of adipocytes?

What about Zeb2 in the bone endothelium? Is it expressed and could be potentially functionally redundant? This point might explain the survival of the constitutive EC-specific mutants.

I am puzzled by the results of the luciferase assays in Fig. 3j and k. While it is easy to appreciate that the activity of the wild-type reporter goes down after the inactivation of Zeb1, it appears that the activity stays high with the mutant construct (right columns in each panel). The reason might be the normalization of the data (control = 100%) and it would be interesting to know whether the activity of the mutant construct is reduced relative to the wild-type construct in absolute terms. If so, the authors should think about a less confusing way to present this data. Alternatively, some explanation is needed if the activity of the mutant construct stays high even in the absence of Zeb1.

The authors have used recombinant Dll4 (r.Dll4) to treat mice and rescue the deficient Notch activity in Zeb1 mutants. While not much detail is provided in the Methods, the website of the supplier states that the recombinant protein covers much of the Dll4 extracellular domain (Ser28-Pro525) and carries a C-terminal His-tag replacing the transmembrane region and the cytoplasmic part of Dll4. Similar constructs (such as biologically more stable Dll4-Fc fusion proteins) have been previously used in the literature and typically interfere with Notch signalling in vitro and in vivo. For example, Noguera-Troise et al. (Nature 2006) saw more endothelial sprouting in Dll4-Fc-treated tumors due to impaired Notch signaling. The basis for this result is that Notch ligands need to be tethered to the plasmamembrane or other surfaces to enable the extraction of the Notch extracellular domain and nuclear translocation of the intracellular region. In this context, the results involving r.Dll4 are very counterintuitive and the underlying mode of action needs to be resolved.

I am not very familiar with the efficiency and specificity of the cationic liposomes used in Fig. 5. In addition, I am not sure whether this method has been used for the bone vasculature before. In any case, the authors should show more convincingly that they can successfully target ECs in the skeletal system with this approach. It would be also critical to know the impact on perivascular cells, some of which also seem to express Zeb1 endogenously (see Fig. 1) and might therefore respond to the liposome treatment.

There is apparently no Supplementary Figure 7.

Minor points:

Is Zeb1 localized in the cell nucleus? In any case, higher magnification images with split channels are needed to show the expression of Zeb1 in vascular vs. perivascular cell populations.

When the authors mention EphB2 on the bottom of page 6, they are presumably thinking about ephrin-B2 (Efnb2).

Reviewer #3:

Remarks to the Author:

The goal of this study was to explore the role of the Zeb1 transcription factor in blood vessel and bone formation in mice. The authors find that deletion of Zeb1 in vascular endothelial cells reduces type H endothelium and bone mass and that this is associated with reduced Notch signaling in this tissue. They also show that ovariectomy in mice reduces Zeb1, type H vessels, and Notch signaling in bone and that delivery of Zeb1 via liposomes prevents these changes as well as bone loss. Based on these findings the authors conclude that therapeutic restoration of Zeb1 activity may be beneficial to patients with low bone mass.

Comments.

1. In figure 1, comparison of Zeb1 abundance in different types of endothelium was accomplished by measuring Zeb1 exclusively by immunofluorescence. This detection method has a low dynamic range. Later in the study, the authors repeatedly analyzed gene expression in L and H type endothelial cells isolated by flow cytometry. A similar approach to compare Zeb1 mRNA in the different cell types would be more convincing.

2. The skeletal analyses of the Zeb1 conditional knockout mice is insufficient. The authors conclude that the low bone mass is due to reduced bone formation. However, they do not measure bone formation. Quantifying immunoreactivity of factors expressed by osteoblasts is not an acceptable surrogate for direct measurement of bone formation by a dual labeling approach. The latter, which is performed by timed injections of fluorescent compounds such as calcein or tetracycline, is the only method to actually measure bone formation. Therefore, the cellular mechanism underlying the low bone mass of the mutant mice remains unclear.

3. Along the same lines, no explanation is provided for the low body weight of the Zeb1 conditional knockout mice. Body weight can have a profound effect on bone mass. One approach that may have circumvented this confounding situation would have been to induce Zeb1 deletion after skeletal growth was complete using the tamoxifen-inducible model. However, this approach was not used. Instead, the authors induced deletion using tamoxifen in young growing mice and analyzed only blood vessels, not bone mass. The authors also did not state whether administration of recombinant Dll4 normalized body weight, as it did bone mass. Therefore, overall, it remains unclear whether Zeb1 expression in blood vessels has a direct effect on bone formation or bone mass.

4. It is unclear why mutation of the E-box binding sites in the Dll4 and Notch1 promoter-reporter constructs did not reduce promoter activity. Zeb1 overexpression stimulated reporter activity indicating that Zeb1 is an activator of these genes. It is puzzling, then, that deletion of the binding sites for an activator had no effect on transcriptional activity of these reporters (figures 3j-k).

5. The results of the ovariectomy experiment presented in figure 5 are highly problematic. A significant issue is the reduced periosteal perimeter in the ovariectomized mice shown in figure 5g. This is an unexpected finding and is inconsistent with numerous published studies. Specifically, ovariectomy of growing mice, which were used in this experiment, leads to increased, not reduced,

periosteal expansion (Journal of Bone and Mineral Research, 25:617–626, 2010). Similarly, if skeletally mature mice are ovariectomized, bone is lost at the endosteum but not the periosteum (Mol Endocrinology 27:649–656, 2013 and Nature Medicine 24:823–833, 2018). Thus, it is unclear how the authors find that ovariectomy of wild type mice resulted in tibiae with a reduced cross-sectional area. Along the same lines, almost 30 years of studies in mice consistently find that ovariectomy increases osteoblast number and bone formation, yet the surrogates measured by the authors suggest reduced osteoblast number (figure 5i). These inconsistencies with abundant published work lead to very low confidence in the author's conclusions regarding the ovariectomy experiment.

6. A minor point is that the introduction refers to "postmenopausal osteoporotic mice". Mice do not undergo a menopause. Ovariectomy of skeletally mature mice is a model of postmenopausal osteoporosis. However, ovariectomy is a surgical procedure and should not be equated with menopause.

Reviewer #4:

Remarks to the Author:

Review Nature Communications: NCOMMS-18-9963423-T

As requested by the editor I have reviewed the manuscript "Endothelial Zeb1 promotes angiogenesis-dependent bone formation and reverses osteoporosis" by Fu et al, specifically regarding the liposomal delivery approach used by the authors in this work.

In their study the author have administered liposomes carrying the Zeb1 gene i.v. into OVX mice. This treatment resulted in increased Zeb1 protein expression in type H vessels and improved vasculature and bone formation compared to empty liposome treated controls. Despite these interesting results, the liposomal delivery of Zeb1 gene, applied in this study, raises some questions and remarks.

The preparation of pcDNA-Zeb1-loaded cationic liposomes as described in the online methods section, lacks information that allows solid interpretation of the liposomal Zeb1 gene delivery in relation to the observed results. 1) There is no physicochemical characterization of the final particle (e.g., size, zeta-potential, dispersity index) nor is there a reference to the characteristics of these liposomes. 2) No information is given on the kinetics of the pcDNA-Zeb1-loaded cationic liposomes and their biodistribution. This is very relevant since the liposomes applied are not 'long-circulating' liposomes which might argue against the targeting of these liposomes to H vessels. Although the authors state that their results indicate efficient targeting of Zeb1 gene delivery, they don't show any data to support this "targeting" other than a 2 fold increase of Zeb1 protein in H vessels. 3) Was there an increase or change in Zeb1 protein expression in other vessels, other cell types and/or organs after treatment with pcDNA-Zeb1-loaded cationic liposomes and to what extend?

As a control for the injected pcDNA-Zeb1-loaded cationic liposomes the authors have administered cationic liposomes (vehicle). An irrelevant pcDNA-loaded cationic liposome would have been a better control since the liposomal particle characteristics would have been the same. The particle characteristics and thus the kinetics of the vehicle will be (very) different from the protamine-cDNA complex loaded cationic liposomes used for Zeb1 gene delivery.

The authors state in their manuscript (results, discussion, supplementary figure 9b) that neither cationic liposomes nor pcDNA-Zeb1-loaded cationic liposomes induce detectable histological alterations, indicating the absence of toxicity. Although I'm not a pathologist, I do seem to see differences in suppl. Fig 9b. I would suggest to have an experienced mouse pathologist to have a

look at these tissues.

Response to Reviewer #1:

In this manuscript, the authors used mice with endothelial cell-specific deletion of Zeb1 to demonstrate that Zeb1 plays an important role in CD31-high, endomucin-high vessel formation in the bone and osteogenesis. Mechanistically, Zeb1 deletion reduces histone acetylation on *Dll4* and *Notch1* promoters, leading to decreased Notch signaling, a critical pathway in angiogenesis and osteogenesis. In addition, add-back of Zeb1 by Zeb1 liposome reversed Zeb's loss-of-function effects on Notch signaling and angiogenesis-dependent osteogenesis. Also, administration of recombinant *Dll4* protein also reversed Zeb1 deletion-induced phenotypes (restored CD31-high EMCN-high angiogenesis and promoted bone formation) in Zeb1 conditional knockout mice. Overall, this is a rigorous study demonstrating the functional importance of Zeb1 in bone angiogenesis and osteogenesis. Here are my comments on the mechanism of Zeb1 in angiogenesis:

1. Figure 3a: it is not clear why the authors specifically focused on the Notch pathway. The Zeb1/miR-200 signaling axis was reported to regulate Notch signaling (EMBO J. 2011 Feb 16; 30(4): 770–782). Does this axis contribute to the role of Zeb1 in angiogenesis?

To investigate the potential mechanisms that endothelial ZEB1 regulates bone angiogenesis, we performed a qPCR screening assay using FACS-sorted bone ECs of ZEB1^{WT} and ZEB1^{ΔEC} mice. We focused on the genes (signal pathways) that are essential for bone angiogenesis and osteogenesis. Importantly, we found that key components of Notch, TGFβ, BMP, PLGF, and FGF1 signaling pathways were substantially reduced in bone ECs of ZEB1^{ΔEC} mice compared to ZEB1^{WT} mice (Fig. 3a; Suppl. Fig. 5d). More importantly, administration of ZEB1^{ΔEC} mice with recombinant *Dll4* protein efficiently restored the impaired Notch, TGFβ, BMP, PLGF, and FGF1 signaling pathways in bone ECs of the mice (Suppl. Fig. 5b, 5d). Also, we observed that Notch signaling was downregulated in the in vitro cultured ZEB1-deleted bone ECs compared to control cells (Suppl. Fig. 5a; Fig. 3g). A recent report has demonstrated that Notch signaling pathway plays a critical role in controlling bone angiogenesis and osteogenesis (Ramasamy et al, Nature, 2014, 507: 376-380). Therefore, we specifically focused on the Notch signaling pathway, and further demonstrate that deletion of endothelial ZEB1 impairs bone angiogenesis and osteogenesis by epigenetically suppressing *Dll4*/Notch signaling.

We have noticed the mentioned paper (Brabletz et al, EMBO J, 2011, 30: 770-782) claiming that ZEB1 indirectly upregulates Notch signaling by reducing miR-200 expression in multiple tumor cell lines. Here we further demonstrate that ZEB1 interacts with histone acetyltransferases (HATs), CREB-binding protein (CBP)/p300 and they co-occupy the promoters of *Dll4* and *Notch1* to DIRECTLY induce their transcriptional activation in bone ECs. At current stage, we could not rule out the possibility that ZEB1/miR-200 signaling axis may also regulate

Notch signaling in bone ECs.

2. Why does Zeb1 deletion specifically affect angiogenesis in the bone but not in other tissues?

An interesting question. In contrast to its robust expression in skeletal endothelium (especially CD31^{hi}EMCN^{hi} endothelium; termed as type H endothelium), ZEB1 expression is largely undetectable in non-skeletal tissues such as spleen, lung, kidney, liver, and heart (Fig. 1a-1d). Constitutive and tamoxifen-inducible deletion of endothelial ZEB1 impairs bone angiogenesis without affecting vessel formation in non-skeletal tissues (Fig. 2a, 2b, 2c; Suppl. Fig. 2a-2g). We propose that the considerably variable consequences of endothelial ZEB1 deletion are due to the highly differential expression of ZEB1 in skeletal endothelium versus non-skeletal endothelium.

3. In addition to Notch signaling, does Zeb1 deletion affect other pathways in CD31-high, endomucin-high endothelial cells? Unbiased profiling (RNA-seq) and pathway analysis should be performed.

As described above, we performed a qPCR screening assay using FACS-sorted bone ECs of ZEB1^{WT} and ZEB1^{ΔEC} mice. We demonstrate that key components of Notch, TGFβ, BMP, PLGF, and FGF1 signaling pathways that are reported to play an essential role in controlling bone angiogenesis and osteogenesis were substantially reduced in ZEB1^{ΔEC} bone ECs as compared to ZEB1^{WT} bone ECs (Suppl. Fig. 5d; Fig. 3a). Notably, we found that key components of TGFβ, BMP, PLGF, and FGF1 signaling pathways were markedly restored in bone ECs of ZEB1^{ΔEC} mice that were treated with recombinant Dll4 protein (Suppl. Fig. 5d). Furthermore, administration of ZEB1^{ΔEC} mice with recombinant Dll4 protein significantly restored the impaired bone angiogenesis and osteogenesis (Fig. 5c-5h). Thus, we have confirmed that Dll4/Notch1 as a major signaling pathway that is actively involved in the regulation of ZEB1 on bone angiogenesis and osteogenesis.

4. Figure 3p and 3q: does knockout of Zeb1 specifically regulate histone acetylation on Dll4 and Notch1 promoters? How about other regions of the genome? Unbiased analysis (ChIP-seq) should be performed.

Snail, another EMT-associated transcription factor, has been reported to function as a transcriptional activator by specifically enhancing histone acetylation but not methylation on the promoters of target genes. Meanwhile, Snail may also act as a transcriptional repressor by specifically increasing histone methylation but not acetylation on the promoter of target genes such as E-cadherin (Hsu et al, Cancer Cell, 2014, 26: 534-548). Here, we have demonstrated that ZEB1, similar to Snail, functions as a transcriptional activator in bone ECs. We found that ZEB1

deletion in bone ECs reduces histone acetylation, including histone 3 lysine 4 acetylation (H3K4Ac), histone 3 lysine 14 acetylation (H3K14Ac), and histone 3 lysine 18 acetylation (H3K18Ac) on the promoters of *Dll4* and *Notch1*, without affecting histone methylation such as histone 3 lysine 27 tri-methylation (H3K27me3, a reported repressive histone mark) on *Dll4* and *Notch1* promoters (**Fig. 4f in revised manuscript**). Taken together, we conclude that ZEB1, an important EMT-associated transcription factor, may also act as a transcriptional activator by specifically enhancing histone acetylation but not methylation on the promoters of target genes.

5. What is the mechanism by which Zeb1 regulates histone acetylation?

A very critical issue! We have performed new experiments to dissect the mechanism by which ZEB1 regulates histone acetylation on the promoters of *Dll4* and *Notch1*. We have found that ZEB1 is physically associated with histone acetyltransferases (HATs), CREB-binding protein (CBP)/p300 in bone ECs (**Fig. 4g**). Sequential ChIP analysis revealed that ZEB1 and CBP/p300 co-occupy the promoters of *Notch1* and *DLL4* in bone ECs (**Fig. 4h**). A promoter luciferase reporter assay was also performed in 293T cells that were transiently transfected with HA-ZEB1, HA-p300 or HA-CBP alone, or ZEB1 in combination with p300 or CBP. The results showed that overexpression of ZEB1, p300 or CBP alone increased *Dll4* and *Notch1* promoter activity by 88/75%, 20/16% or 23/15%, respectively, while overexpression of ZEB1 in combination with p300 or CBP leads to a 6.0/4.1-fold or 7.0/5.4-fold increase in *Dll4* and *Notch1* promoter activity, respectively (**Fig. 4i and 4j**). These findings suggest that CBP/p300 interact with ZEB1 in bone ECs and they co-occupy the promoters of *Dll4* and *Notch1* to enhance histone acetylation on the promoters (**Fig. 4f**), thus inducing transcriptional activation of *Dll4* and *Notch1* (**Fig. 4c**).

Response to Reviewer #2:

In their manuscript "Endothelial Zeb1 promotes angiogenesis-dependent bone formation and reverses osteoporosis", Fu et al. report that the zinc-finger transcription factor Zeb1 controls Notch signaling in bone endothelial cells (ECs) and is thereby critical for the coupling of angiogenesis and osteogenesis. The authors have generated and characterized EC-specific Zeb1 mutant mice, and their data convincingly shows that both vessel growth and osteogenesis in long bone are impaired relative to littermate controls. At the molecular level, loss of Zeb1 leads to reduced expression of Notch pathway genes and other data support that this is due to binding of Zeb1 to various E-box motifs in the Dll4 and Notch1 promoters. The authors also show that treatment of Zeb1 mutant mice with a soluble, recombinant version of Dll4 restores Notch signaling and improves osteogenesis. A final set of experiments indicates that Zeb1 expression is reduced in osteoporotic patients and in the ovariectomy mouse model. Administration of Zeb1-packaged liposomes increases both the number of osteoprogenitor cells and the amount mineralized bone in ovariectomized mice, suggesting that Zeb1 might be a promising therapeutic target.

Previous work by other has already shown that Zeb1 is an important regulator of craniofacial and skeletal development, but it was unclear whether the protein plays a role in the endothelium or in other cell types in bone. There are also reports about positive regulation of VEGF-A expression and Notch signaling by Zeb1, which are obviously relevant for this study and should be at least mentioned in the Discussion.

We mentioned these relevant reports in the Results and Discussion (highlighted in red in the revised manuscript).

Overall, the data is mostly of good quality and it is easy to follow the logic of the manuscript. There are, however, a couple of important issues that need to be addressed prior to publication:

I could not find the protocol for the tamoxifen administration anywhere in the manuscript. The wording of the third paragraph in the Online Methods is rather confusing. Presumably, mutants generated with the Tie2-Cre line were not treated with tamoxifen. It is also not stated whether control mice in the experiments with the Cdh5-CreERT2 line were also treated with tamoxifen. These issues should be clarified.

We have moved the corrected online methods to text, and all detailed information related to tamoxifen treatment was also included. As described in the method, Cdh5-CreERT2⁺;ZEB1^{fl/fl} and Cdh5-CreERT2⁺;ZEB1^{fl/fl} mice were injected with tamoxifen to generate ZEB1^{WT} and ZEB1^{iAEC} mice, respectively.

I am concerned about the specificity of the Dll4 immunostaining in Fig. 3b. Instead of the expected EC-specific staining, a lot of signal can be seen in perivascular cell

populations. This, the alterations in the mutant samples might simply reflect a reduction of perivascular cells but not necessarily impaired Dll4 expression. This a critical issue with relevance for the main conclusions of the study.

Given this criticism, we reperformed all Dll4 staining experiments using an alternative donkey serum (in preparation of blocking buffer) from another vendor (Jackson, #017000121). The new data are now more convincing. As shown, Dll4 is primarily expressed in EMCN⁺ bone ECs with little found in perivascular cells (Fig. 3b, 3c, left panels; Fig. 6g, top panels; Suppl. Fig. 5b, top panels).

It is noteworthy that the cellularity of the bone marrow is strongly reduced in EC-specific Zeb1 mutants (see Suppl. Fig. 4b; bottom panels). What is the cause for this defect? Is there a higher abundance of adipocytes?

My technician accidentally picked up scratched tissue sections for H.&E. staining. These damaged sections were now replaced with intact ones. As shown, bone volume and trabecular number were remarkably reduced in tibia of 50-week-old ZEB1^{ΔEC} mice compared to ZEB1^{WT} mice (Suppl. Fig. 3d, right panels). Also, we failed to observe a visible change in bone marrow between ZEB1^{ΔEC} and ZEB1^{WT} tibia (Suppl. Fig. 3d, right panels).

What about Zeb2 in the bone endothelium? Is it expressed and could be potentially functionally redundant? This point might explain the survival of the constitutive EC-specific mutants.

A qPCR analysis of FACS-sorted bone ECs of ZEB1^{WT} and ZEB1^{ΔEC} mice revealed that ZEB2 transcript levels in type H bone ECs were significantly lower than in type L bone ECs, an expression pattern that is highly distinct from ZEB1 in bone ECs (Fig. 1e). Intriguingly, we further demonstrated that ZEB2 expression levels in bone ECs were significantly decreased following ZEB1 excision both in vivo and in vitro (Fig. 3a, 3g). These results suggest that ZEB1 deletion in bone ECs does not cause a compensatory increase in ZEB2 expression but induces a marked reduction in ZEB2 expression.

I am puzzled by the results of the luciferase assays in Fig. 3j and k. While it is easy to appreciate that the activity of the wild-type reporter goes down after the inactivation of Zeb1, it appears that the activity stays high with the mutant construct (right columns in each panel). The reason might be the normalization of the data (control = 100%) and it would be interesting to know whether the activity of the mutant construct is reduced relative to the wild-type construct in absolute terms. If so, the authors should think about a less confusing way to present this data. Alternatively, some explanation is needed if the activity of the mutant construct stays high even in the absence of Zeb1.

To precisely compare WT versus E-box mutated (MUT) promoter activity in control and ZEB1-deleted bone ECs, all data were normalized to control (luciferase activity of WT reporter construct in control cells is set to 1). As predicted, in control cells, luciferase activity of MUT Dll4 or Notch1 reporter constructs was significantly lower than corresponding WT reporter constructs (Fig. 4b, left and right panels, compare column 3 and 1). ZEB1-deleted bone ECs markedly decreased luciferase activity of WT reporter constructs (Fig. 4b, left and right panels, compare column 2 and 1), without affecting luciferase activity of MUT reporter constructs (Fig. 4b, left and right panels, compare column 4 and 3).

The authors have used recombinant Dll4 (r.Dll4) to treat mice and rescue the deficient Notch activity in Zeb1 mutants. While not much detail is provided in the Methods, the website of the supplier states that the recombinant protein covers much of the Dll4 extracellular domain (Ser28-Pro525) and carries a C-terminal His-tag replacing the transmembrane region and the cytoplasmic part of Dll4. Similar constructs (such as biologically more stable Dll4-Fc fusion proteins) have been previously used in the literature and typically interfere with Notch signaling in vitro and in vivo. For example, Noguera-Troise et al. (Nature 2006) saw more endothelial sprouting in Dll4-Fc-treated tumors due to impaired Notch signaling. The basis for this result is that Notch ligands need to be tethered to the plasmamembrane or other surfaces to enable the extraction of the Notch extracellular domain and nuclear translocation of the intracellular region. In this context, the results involving r.Dll4 are very counterintuitive and the underlying mode of action needs to be resolved.

As reported by Carlson et al. (Nature, 2008, 454: 528-532) and Hu et al. (J Exp Med, 2014, 211: 281-295), muscle satellite cells and ECs seeded on dishes pre-coated recombinant Dll4 (R&D, #1389) significantly stimulated Notch activity, as assessed by NICD1/Notch1 immunoblotting and immunofluorescent staining. In our study, we performed a similar experiment using the exactly same recombinant Dll4 protein (R&D, #1389). We found that ZEB1-deleted bone ECs seeded on r.Dll4 pre-coated culture dishes (or chamber wells) remarkably restored the impaired Notch signaling, as assessed by NICD1 immunoblotting and immunofluorescent staining (Fig. 5a, 5b) and qPCR (for *Dll4*, *Notch1*, *Hes1*, *Hey1*, and *Efnb2* transcripts; Suppl. Fig. 5a). More importantly, we found that administration of ZEB1^{ΔEC} mice with r.Dll4 protein efficiently rescued the deficient Notch signaling, thus recovering the impaired bone angiogenesis and osteogenesis (Suppl. Fig. 5b, Fig. 5c-5h).

I am not very familiar with the efficiency and specificity of the cationic liposomes used in Fig. 5. In addition, I am not sure whether this method has been used for the bone vasculature before. In any case, the authors should show more convincingly that they can successfully target ECs in the skeletal system with this approach. It would be also critical to know the impact on perivascular cells, some of which also seem to express

Zeb1 endogenously (see Fig. 1) and might therefore respond to the liposome treatment.

To better evaluate the efficiency and specificity of cationic liposome-delivered *ZEB1* gene in vivo, we constructed a new pcDNA3.1+C-eGFP-ZEB1 vector which expresses recombinant C-eGFP fused ZEB1 protein in the target cells and packaged the vector (or the backbone vector pcDNA3.1+C-eGFP) into cationic liposomes. Eight-week-old Sham and OVX mice were intravenously injected with pcDNA3.1+C-eGFP-packaged liposomes (designed Lipo.-Vector-GFP) or pcDNA3.1+C-eGFP-ZEB1-packaged liposome (designed Lipo.-ZEB1-GFP) at a dose of 4 µg twice a week for 6 consecutive weeks. Tibia and non-skeletal tissues such as lung, liver, spleen, and kidney were dissected, sectioned and subjected to immunofluorescent and histological analyses. Notably, GFP immunofluorescent analysis revealed that liposome-delivered Vector-GFP and ZEB1-GFP plasmids were predominantly expressed in the metaphysis of tibia (Fig. 6c) with remarkably lower expressions in the diaphysis of tibia and non-skeletal tissues such as lung, liver, and spleen (Suppl. Fig. 7c). Intriguingly, a large proportion of GFP⁺ cells was identified as EMCN⁺ bone ECs, indicating that bone ECs in the metaphysis of tibia are the major target cells of Lipo.-Vector-GFP and Lipo.-ZEB1-GFP (Fig. 6c). However, a small proportion of GFP⁺ cells can be identified as EMCN⁻ perivascular cells that may also respond to the liposome treatment (Fig. 6c). In metaphyseal cells of tibia, Vector-GFP recombinant protein was detected in cytoplasmic compartment (in green; Fig. 6c, left and middle panels), while ZEB1-GFP recombinant proteins were detected in nuclear compartment (in yellow; Fig. 6c, right panel). In the metaphysis of tibia of OVX + Lipo.-ZEB1-GFP mice (Fig. 6c, right panel), the positivity of endogenous ZEB1⁺ bone ECs (i.e. the ratio of red/white: white) and exogenous ZEB1⁺ bone ECs (i.e. the ratio of yellow/white: white) was 13.4% and 28.3%, respectively (Fig. 6c, right panel; Fig. 6d, left panel), suggesting that exogenous recombinant ZEB1 protein was efficiently delivered to and expressed in bone ECs. Accordingly, ZEB1 expression levels in bone ECs of OVX + Vector-GFP mice were decreased by 60% as compared to Sham + Vector-GFP mice, and the levels of ZEB1 (including endogenously- and exogenously-expressed ZEB1) in bone ECs of OVX + Lipo.-ZEB1-GFP mice were restored to ~ 90% those of Sham mice (Fig. 6c; Fig. 6d, right panel).

There is apparently no Supplementary Figure 7.

The original Suppl. Fig. 7 depicting Dll4 and Notch1 immunofluorescent staining in ZEB1^{WT} and ZEB1^{iΔEC} mice (i.e. control and inducible endothelial ZEB1-deleted mice) was moved to text (Fig. 3c, 3e in revised version).

Minor points:

Is Zeb1 localized in the cell nucleus? In any case, higher magnification images with

split channels are needed to show the expression of Zeb1 in vascular vs. perivascular cell populations.

ZEB1 is widely recognized as a nuclear protein. We have presented higher magnification images showing that ZEB1 is indeed localized in nuclear compartment of both ECs and perivascular cells in the metaphysis of tibia (Suppl. Fig. 1d, left panels; 1f, left panels). Furthermore, we have demonstrated that ZEB1 expressions were specifically deleted in ECs but not perivascular cells in ZEB1^{ΔEC} and ZEB1^{iΔEC} mice (Suppl. Fig. 1d, left panels; 1f, right panels).

When the authors mention EphB2 on the bottom of page 6, they are presumably thinking about ephrin-B2 (Efnb2).

Corrected.

Response to Reviewer #3:

The goal of this study was to explore the role of the Zeb1 transcription factor in blood vessel and bone formation in mice. The authors find that deletion of Zeb1 in vascular endothelial cells reduces type H endothelium and bone mass and that this is associated with reduced Notch signaling in this tissue. They also show that ovariectomy in mice reduces Zeb1, type H vessels, and Notch signaling in bone and that delivery of Zeb1 via liposomes prevents these changes as well as bone loss. Based on these findings the authors conclude that therapeutic restoration of Zeb1 activity may be beneficial to patients with low bone mass.

Comments.

1. In figure 1, comparison of Zeb1 abundance in different types of endothelium was accomplished by measuring Zeb1 exclusively by immunofluorescence. This detection method has a low dynamic range. Later in the study, the authors repeatedly analyzed gene expression in L and H type endothelial cells isolated by flow cytometry. A similar approach to compare Zeb1 mRNA in the different cell types would be more convincing.

As requested, we performed RT-qPCR analysis on FACS-sorted type H (EMCN^{hi}CD31^{hi}) and type L (EMCN^{lo}CD31^{lo}) bone ECs of tibia. The data consistently confirmed a predominant expression of ZEB1 in type H bone ECs with a weak expression in type L bone ECs (Fig. 1e, left two columns). In contrast, ZEB2 expression levels in type H bone ECs were substantially lower than in type L bone ECs (Fig. 1e, right two columns).

2. The skeletal analyses of the Zeb1 conditional knockout mice is insufficient. The authors conclude that the low bone mass is due to reduced bone formation. However, they do not measure bone formation. Quantifying immunoreactivity of factors expressed by osteoblasts is not an acceptable surrogate for direct measurement of bone formation by a dual labeling approach. The latter, which is performed by timed injections of fluorescent compounds such as calcein or tetracycline, is the only method to actually measure bone formation. Therefore, the cellular mechanism underlying the low bone mass of the mutant mice remains unclear.

We thank the reviewer for this recommendation. As suggested, we performed in vivo double calcein labeling experiments and confirmed that ZEB1^{AEC} mice had decreased bone formation rates relative to their littermate controls (Fig. 2o, 2p).

3. Along the same lines, no explanation is provided for the low body weight of the Zeb1 conditional knockout mice. Body weight can have a profound effect on bone

mass. One approach that may have circumvented this confounding situation would have been to induce Zeb1 deletion after skeletal growth was complete using the tamoxifen-inducible model. However, this approach was not used. Instead, the authors induced deletion using tamoxifen in young growing mice and analyzed only blood vessels, not bone mass. The authors also did not state whether administration of recombinant Dll4 normalized body weight, as it did bone mass. Therefore, overall, it remains unclear whether Zeb1 expression in blood vessels has a direct effect on bone formation or bone mass.

In response to this criticism, we intraperitoneally injected 7-week-old adult Cdh5-CreERT2⁻;ZEB1^{fl/fl} and Cdh5-CreERT2⁺;ZEB1^{fl/fl} mice with tamoxifen (100 μ l, 10 mg/ml in corn oil) every other day for 2 consecutive weeks to generate ZEB1^{WT} and ZEB1 ^{Δ EC} mice, respectively, and then measured their body weights, bone formation and bone angiogenesis at 10 weeks of ages. We demonstrated that tamoxifen-inducible endothelial ZEB1 deletion in adult mice had no impact on body weights (Suppl. Fig. 3f, right two columns), while indeed resulted in reduced bone angiogenesis (Suppl. Fig. 2e, 2f) in tandem with remarkably impaired bone formation (Suppl. Fig. 3i, 3j). As requested, we performed micro-CT assays of tibia in ZEB1^{WT} and ZEB1 ^{Δ EC} young growing mice (8-day-old mice were intraperitoneally injected with 0.1 mg tamoxifen every day for 7 consecutive days, and tissues were dissected at 3 weeks of ages). We demonstrated that tamoxifen-inducible endothelial ZEB1 deletion in young growing mice did lower body weights in the mice (Suppl. Fig. 3f, left two columns), and ZEB1 ^{Δ EC} mice exhibited reduced bone formation (Suppl. Fig. 3g, 3h). In addition, we also found that administration of ZEB1 ^{Δ EC} mice with recombinant Dll4 protein efficiently recovered body weights in the mice (Suppl. Fig. 5c, compare column 4 and 3).

4. It is unclear why mutation of the E-box binding sites in the Dll4 and Notch1 promoter-reporter constructs did not reduce promoter activity. Zeb1 overexpression stimulated reporter activity indicating that Zeb1 is an activator of these genes. It is puzzling, then, that deletion of the binding sites for an activator had no effect on transcriptional activity of these reporters (figures 3j-k).

This is due to the normalization of the data (the luciferase activity of MUT reporter construct in control bone ECs was set to 1). To precisely compare WT versus E-box mutated (MUT) promoter activity in control and ZEB1-deleted bone ECs, all data were normalized to control (luciferase activity of WT reporter construct in control cells is set to 1). As predicted, luciferase activity of MUT Dll4 and Notch1 reporter constructs was significantly reduced in control cells as compared to corresponding WT reporter constructs (Fig. 4b, left and right panels, compare column 3 and 1). ZEB1-deleted bone ECs markedly decreased luciferase activity of WT reporter constructs (Fig. 4b, left and right panels, compare column 2 and 1), without affecting luciferase activity of MUT reporter constructs (Fig. 4b, left and right panels, compare column 4 and 3).

5. The results of the ovariectomy experiment presented in figure 5 are highly problematic. A significant issue is the reduced periosteal perimeter in the ovariectomized mice shown in figure 5g. This is an unexpected finding and is inconsistent with numerous published studies. Specifically, ovariectomy of growing mice, which were used in this experiment, leads to increased, not reduced, periosteal expansion (Journal of Bone and Mineral Research, 25:617–626, 2010). Similarly, if skeletally mature mice are ovariectomized, bone is lost at the endosteum but not the periosteum (Mol Endocrinology 27:649–656, 2013 and Nature Medicine 24:823–833, 2018). Thus, it is unclear how the authors find that ovariectomy of wild type mice resulted in tibiae with a reduced cross-sectional area. Along the same lines, almost 30 years of studies in mice consistently find that ovariectomy increases osteoblast number and bone formation, yet the surrogates measured by the authors suggest reduced osteoblast number (figure 5i). These inconsistencies with abundant published work lead to very low confidence in the author's conclusions regarding the ovariectomy experiment.

Thanks for pointing out this issue. The facility technician in micro-CT core made a mistake of handling image scaling across different samples. In response to another reviewer's criticisms, we decided to reperform the whole OVX experiments as described in the original Fig. 5 (Fig. 5 was split into Fig. 6 and Fig. 7 in the revised manuscript). We constructed a new pcDNA3.1+C-eGFP-ZEB1 vector which expresses recombinant C-eGFP fused ZEB1 protein in the target cells and packaged the vector (or the backbone vector pcDNA3.1+C-eGFP) into cationic liposomes. Eight-week-old Sham and OVX mice were intravenously injected with pcDNA3.1+C-eGFP-packaged liposomes (designed Lipo.-Vector-GFP) or pcDNA3.1+C-eGFP-ZEB1-packaged liposome (designed Lipo.-ZEB1-GFP) at a dose of 4 μ g twice a week for 6 consecutive weeks. The tibias were dissected and subjected to micro-CT, immunofluorescent, and histological analyses. As predicted, micro-CT analysis revealed that tibia of OVX mice exhibited increased cross-sectional endosteal perimeters (might due to bone loss at the endosteum) in tandem with remarkably decreased cortical thickness (Ct. Th.), trabecular bone density (Tb. BV/TV), trabecular number (Tb. Nb.), and trabecular thickness (Tb. Th.) (Fig. 7a, left and middle panels; 7b, left and middle columns). Notably, administration of OVX mice with Lipo.-ZEB1-GFP efficiently restored the impaired bone formation in the mice (Fig. 7a, middle and right panels; 7b, middle and right columns).

Recently, Kusumbe et al (Nature, 2014, 507: 323-328) reported that Osterix-expressing osteoprogenitors were predominantly positioned around type H vessels in the metaphysis of tibia. In the present study, we further demonstrated that tibia of OVX mice displayed markedly reduced type H vessel formation (Fig. 6f, compare left and middle columns) in tandem with decreased numbers of Runx2⁺ pre-osteoprogenitors and Osterix⁺ osteoprogenitors (Fig. 7c, compare left and middle panels; 7d, compare left and middle columns). These results are consistent with previous reports demonstrating that numbers of Runx2⁺ and

Osterix⁺ cells are reduced in the bone of OVX mice compared to Sham mice (Mora-Raimundo, ACS Nano, 2019, 13: 5451-5464; Yang L, Biochem Biophys Res Commun, 2018, 504: 941-948; Pan et al, Med Chem Commun, 2018, 9: 1359-1368; Jing H et al, Cell Death Dis, 2018, 9:176; Wang et al, Stem Cell Res Ther, 2018, 9: 65; Jiang et al, Cell Tissue Res, 2017, 367: 257-267).

6. A minor point is that the introduction refers to “postmenopausal osteoporotic mice”. Mice do not undergo a menopause. Ovariectomy of skeletally mature mice is a model of postmenopausal osteoporosis. However, ovariectomy is a surgical procedure and should not be equated with menopause.

Corrected. We replaced “postmenopausal osteoporotic mice” with “ovariectomized mice”.

Response to Reviewer #4:

Review Nature Communications: NCOMMS-18-9963423-T

As requested by the editor I have reviewed the manuscript “Endothelial Zeb1 promotes angiogenesis-dependent bone formation and reverses osteoporosis” by Fu et al, specifically regarding the liposomal delivery approach used by the authors in this work.

In their study the authors have administered liposomes carrying the Zeb1 gene i.v. into OVX mice. This treatment resulted in increased Zeb1 protein expression in type H vessels and improved vasculature and bone formation compared to empty liposome treated controls. Despite these interesting results, the liposomal delivery of Zeb1 gene, applied in this study, raises some questions and remarks.

The preparation of pcDNA-Zeb1-loaded cationic liposomes as described in the online methods section, lacks information that allows solid interpretation of the liposomal Zeb1 gene delivery in relation to the observed results. 1) There is no physicochemical characterization of the final particle (e.g., size, zeta-potential, dispersity index) nor is there a reference to the characteristics of these liposomes. 2) No information is given on the kinetics of the pcDNA-Zeb1-loaded cationic liposomes and their biodistribution. This is very relevant since the liposomes applied are not ‘long-circulating’ liposomes which might argue against the targeting of these liposomes to H vessels. Although the authors state that their results indicate efficient targeting of Zeb1 gene delivery, they don’t show any data to support this “targeting” other than a 2-fold increase of Zeb1 protein in H vessels. 3) Was there an increase or change in Zeb1 protein expression in other vessels, other cell types and/or organs after treatment with pcDNA-Zeb1-loaded cationic liposomes and to what extend?

To directly assess the efficiency and specificity of cationic liposome-delivered ZEB1 gene *in vivo*, we constructed a new pcDNA3.1+C-eGFP-ZEB1 vector that can express recombinant eGFP-fused ZEB1 protein in the target cells and packaged the vector (or the backbone vector pcDNA3.1+C-eGFP) into cationic liposomes. The detailed information of preparation of pcDNA3.1+C-eGFP-packaged liposomes (designed Lipo.-Vector-GFP) and pcDNA3.1+C-eGFP-ZEB1-packaged liposome (designed Lipo.-ZEB1-GFP) was included in the Method that was moved to the text in the revised manuscript.

1) For preparation of cationic liposomes, 1,2-dioleoyl-*sn*-glycero-3-phosphoethanolamine (DOPE), 1,2-dioleoyl-3-trimethylammonium propane (DOTAP) and cholesterol were dissolved in chloroform at a molar ratio of 4:3:3, evaporated in a rotary evaporator at 40 °C for 30 min, dried overnight, hydrated with dd H₂O at 40 °C for 20 min, and finally ultrasounded at an intensity of 15%. Vector-GFP (0.2 mg/ml) or ZEB1-GFP (0.2 mg/ml) was mixed with cationic protamine at different N:P ratios (the molar ratio of nitrogen in protamine to phosphate in DNA vector) to form the nano-sized

DNA/protamine complex (designed Protamine-Vector-GFP and Protamine-ZEB1-GFP). DNA packing efficiency was examined by agarose gel electrophoresis. The results showed that protamine can completely condense Vector-GFP and ZEB1-GFP at the N:P ratio of 3:1 and 4:1, respectively (**Supplementary Fig. 6a**). The resulting DNA/protamine complex was subsequently mixed with cationic liposomes at 60 nmol lipid/ μ g DNA to generate Lipo.-Vector-GFP and Lipo.-ZEB1-GFP, respectively. Agarose gel electrophoresis analysis showed that Vector-GFP and ZEB1-GFP were completely encapsulated in the cationic liposomes under these conditions (**Supplementary Fig. 6b**). Particle size and zeta potential were measured using a Zetasizer NANO ZSP (Malvern Panalytical). The final Lipo.-Vector-GFP had a particle size of 151.2 ± 5.7 nm (mean \pm s.d.), a zeta potential of $+24.9 \pm 1.6$ mv, and a polydispersity index (PDI) of 0.286 ± 0.017 , respectively, while the final Lipo.-ZEB1-GFP had 147.5 ± 4.9 nm, $+23.8 \pm 2.1$ mv, and 0.270 ± 0.017 , respectively (**Suppl. Fig. 6c**), confirming largely comparable physicochemical characteristics of Lipo.-Vector-GFP versus Lipo.-ZEB1-GFP.

- 2) For pharmacokinetic analysis, pcDNA3.1+C-eGFP control vector or pcDNA3.1+C-eGFP-ZEB1 vector was packaged into rhodamine PE labelled liposome, and the DNA-packaged liposomes (designed Rho.-Lipo.-Vector-GFP and Rho.-Lipo.-ZEB1-GFP, respectively) were intravenously injected into Sprague Dawley rats at a dose of 0.8 mg/kg; The blood samples collected at 0, 0.125, 0.25, 0.5, 1, 2, 4, 8, 12, and 24 hours post treatment were centrifuged, and the supernatant plasma was collected for fluorescence intensity measurement using a SpectraMax iD5 Multi-Mode Microplate Reader (Molecular Devices). As shown, the fluorescence intensity-time curves of Rho.-Lipo.-Vector-GFP and Rho.-Lipo.-ZEB1-GFP were largely comparable (**Suppl. Fig. 6d**). Then, the pharmacokinetic parameters of Rho.-Lipo.-Vector-GFP and Rho.-Lipo.-ZEB1-GFP were calculated to quantitatively evaluate the pharmacokinetic properties. As shown, elimination half-life ($t_{1/2}$, h) of Rho.-Lipo.-Vector-GFP and Rho.-Lipo.-ZEB1-GFP was 4.74 ± 0.40 and 4.63 ± 0.31 , respectively; Maximal plasma concentration (C_{max} , $\times 10^4$ a.u./ml) of Rho.-Lipo.-Vector-GFP and Rho.-Lipo.-ZEB1-GFP was 13.91 ± 1.00 and 13.42 ± 1.49 , respectively; Area under the plasma concentration-time curve ($AUC_{0-\infty}$, $\times 10^4$ a.u./ml \times h) of Rho.-Lipo.-Vector-GFP and Rho.-Lipo.-ZEB1-GFP was 64.03 ± 5.18 and 57.95 ± 4.61 , respectively; Mean residence time (MRT, h) of Rho.-Lipo.-Vector-GFP and Rho.-Lipo.-ZEB1-GFP was 6.21 ± 0.19 and 6.05 ± 0.14 , respectively (**Suppl. Fig. 6e**). These data confirmed comparable pharmacokinetic properties of Rho.-Lipo.-Vector-GFP versus Rho.-Lipo.-ZEB1-GFP. For biodistribution analysis, pcDNA3.1+C-eGFP-ZEB1 vector was packaged into 1,1'-Di-octadecyl-3,3,3',3'-tetramethylindotricarbocyanine iodide (DiR)-labelled liposome, and the DNA-packaged liposomes (designed DiR-Lipo.-ZEB1-GFP) were intravenously injected into 8-week-old C57BL6 mice at a dose of 1.0 mg/kg; hind limbs, kidneys, spleen, lungs, brain, heart, and liver were dissected 24 h post injection and subjected to biophotonic

imaging assay for evaluation of organ distribution of DiR-Lipo.-ZEB1-GFP. As shown, the DiR signal was readily detected in hind limbs, liver, spleen, lung, and kidneys, while the signal was hardly undetectable in brain and heart (**Suppl. Fig. 6f, 6g**), validating efficient distribution of DiR-Lipo.-ZEB1-GFP to the skeletal and non-skeletal organs.

- 3) We further evaluated EXPRESSIONS of liposome-delivered pcDNA3.1+C-eGFP (Vector-GFP) and pcDNA3.1+C-eGFP-ZEB1 (ZEB1-GFP) plasmids in target cells using an OVX mouse model. Eight-week-old Sham and OVX mice were intravenously injected with Vector-GFP-packaged liposomes (designed Lipo.-Vector-GFP) or ZEB1-GFP-packaged liposome (designed Lipo.-ZEB1-GFP) at 4 µg per mouse twice a week for 6 consecutive weeks. Tibia and non-skeletal tissues such as lung, liver, spleen, and kidney were dissected, sectioned and subjected to immunofluorescent and histological analyses. Notably, GFP immunofluorescent analysis revealed that liposome-delivered Vector-GFP and ZEB1-GFP plasmids were predominantly expressed in the metaphysis of tibia (**Fig. 6c**) with remarkably lower expressions in the diaphysis of tibia and non-skeletal tissues such as lung, liver, and spleen (**Suppl. Fig. 7c**). A large proportion of GFP⁺ cells was identified as EMCN⁺ bone ECs, indicating that bone ECs in the metaphysis of tibia are the major target cells of Lipo.-Vector-GFP and Lipo.-ZEB1-GFP (**Fig. 6c**). However, a small proportion of GFP⁺ cells can be identified as EMCN⁻ perivascular cells that may also respond to the liposome treatment (**Fig. 6c**). Furthermore, in the metaphysis of tibia, Vector-GFP and ZEB1-GFP recombinant proteins were detected in cytoplasm (in green) and nucleus (in yellow) of metaphyseal cells, respectively (**Fig. 6c**). In the metaphysis of tibia of OVX + Lipo.-ZEB1-GFP mice (**Fig. 6c, right panels**), the positivity of endogenous ZEB1⁺ bone ECs (i.e. the ratio of red/white: white) and exogenous ZEB1⁺ bone ECs (i.e. the ratio of yellow/white: white) was 13.4% and 28.3%, respectively (**Fig. 6c, right panels; Fig. 6d, right panels**), suggesting that exogenous recombinant ZEB1 protein was efficiently delivered to and expressed in bone ECs. Accordingly, ZEB1 expression levels in bone ECs of OVX + Vector-GFP mice were decreased by 60% as compared to Sham + Vector-GFP mice, and the levels of ZEB1 (including endogenously- and exogenously-expressed ZEB1) in bone ECs of OVX + Lipo.-ZEB1-GFP mice were restored to ~ 90% those of Sham mice (**Fig. 6c; Fig. 6d, left panels**). Most importantly, we observed that bone angiogenesis and osteogenesis were both impaired in OVX mice which was efficiently restored following Lipo.-ZEB1-GFP treatment (**Fig. 6e, 6f, 7a-7d**). At current stage, we don't know why liposome-delivered ZEB1-GFP (or Vector-GFP) plasmids were predominantly expressed in metaphyseal cells (mostly in type H bone ECs) of tibia. However, one possible explanation is that metaphyseal cells with highly metabolic activity (Kusumbe et al, Nature, 2014, 507: 323-328) are more prone to uptake and express ZEB1-GFP (or Vector-GFP) plasmids than cells in the diaphysis of tibia and other non-skeletal tissues.

As a control for the injected pcDNA-Zeb1-loaded cationic liposomes the authors have administered cationic liposomes (vehicle). An irrelevant pcDNA-loaded cationic liposome would have been a better control since the liposomal particle characteristics would have been the same. The particle characteristics and thus the kinetics of the vehicle will be (very) different from the protamine-cDNA complex loaded cationic liposomes used for Zeb1 gene delivery.

In response to this criticism, we constructed a new pcDNA3.1+C-eGFP-ZEB1 vector and packaged the vector (or the backbone vector pcDNA3.1+C-eGFP) into cationic liposomes. As shown, physicochemical characteristics and pharmacokinetics of Lipo.-Vector-GFP and Lipo.-ZEB1-GFP are largely comparable (see above for detailed information; Suppl. Fig. 6a-6g).

The authors state in their manuscript (results, discussion, supplementary figure 9b) that neither cationic liposomes nor pcDNA-Zeb1-loaded cationic liposomes induce detectable histological alterations, indicating the absence of toxicity. Although I'm not a pathologist, I do seem to see differences in suppl. Fig 9b. I would suggest to have an experienced mouse pathologist to have a look at these tissues.

We reperformed the whole OVX experiments, and the H.&E.-stained sections were presented to an experienced mouse pathologist. The pathologist has confirmed that there are no detectable histological alterations in the vital organs of liposome-treated mice (Suppl. Fig. 7d in the revised manuscript).

Reviewers' Comments:

Reviewer #1:

Remarks to the Author:

The authors addressed my previous points. I have no additional concerns.

Reviewer #2:

Remarks to the Author:

The authors have made extensive revisions and addressed all my questions. There are still a few typographic errors, which are presumably dealt with by the copy editor. Apart from this, the manuscript is now suitable for publication.

Reviewer #3:

Remarks to the Author:

The authors have adequately addressed the concerns of my previous review.

Reviewer #4:

Remarks to the Author:

The points I have raised reviewing this manuscript have been addressed convincingly by the authors in their revised manuscript.

The authors have performed sound experiments to do so and included all relevant information in their manuscript.

REVIEWERS' COMMENTS:

Reviewer #1 (Remarks to the Author):

The authors addressed my previous points. I have no additional concerns.

Thanks!

Reviewer #2 (Remarks to the Author):

The authors have made extensive revisions and addressed all my questions. There are still a few typographic errors, which are presumably dealt with by the copy editor. Apart from this, the manuscript is now suitable for publication.

Thanks! We will correct these typographic errors.

Reviewer #3 (Remarks to the Author):

The authors have adequately addressed the concerns of my previous review.

Thanks!

Reviewer #4 (Remarks to the Author):

The points I have raised reviewing this manuscript have been addressed convincingly by the authors in their revised manuscript. The authors have performed sound experiments to do so and included all relevant information in their manuscript.

Thanks!